

# Evaluation of WRF 4.5.1 surface layer scheme representation of temperature inversions over boreal forests

Julia Maillard[1], Jean-Christophe Raut[1], and François Ravetta[1]

[1]LATMOS/IPSL, Sorbonne Université, UVSQ, CNRS, 4 place Jussieu, 75005 Paris, France

**Correspondence:** Jean-Christophe Raut (jean-christophe.raut@latmos.ipsl.fr)

**Abstract.** In this study, the Noah Land Surface Model used in conjunction with the Mellor-Janjić-Yamada surface layer scheme (hereafter, Noah-MYJ) and the Noah MultiPhysics scheme (Noah-MP) from the WRF 4.5.1 meso-scale model are evaluated with regards to their performance in reproducing positive temperature gradients over forested areas in the Arctic winter. First, simplified versions of the WRF schemes, recoded in Python, are compared with conceptual models of the surface layer in order to gain insight into the dependence of the temperature gradient on the wind speed at the top of the surface layer. It is shown that the WRF schemes place strong limits on the turbulent collapse, leading to lower surface temperature gradient at low wind speeds than in the conceptual models. We implemented modifications to the WRF schemes to correct this effect. The original and modified versions of Noah-MYJ and Noah-MP are then evaluated compared to long-term measurements at the Ameriflux Poker Flats Research Range, a forest site in Interior Alaska. Noah-MP is found to perform better than Noah-MYJ because the former is a 2-layer model which explicitly takes into account the effect of the forest canopy. Indeed a non-negligible temperature gradient is maintained below the canopy at high wind speeds, leading to overall larger gradients than in the absence of vegetation. Furthermore, the modified versions are found to perform better than the original versions of each scheme because they better reproduce strong temperature gradients at low wind speeds.

## 1 Introduction

Surface-based temperature inversions (SBIs) are extremely frequent in the cold, dark conditions of the Arctic winter (Serreze et al., 1992; Bradley et al., 1992). The usual pattern is that cloudy conditions are associated to a near-neutral surface layer, while clear skies are associated to strong SBIs (Malingowski et al., 2014). However, modelling temperature inversions remains a challenge and an area of ongoing study (Steeneveld et al., 2006; Sterk et al., 2013; Holtslag et al., 2013; Baas et al., 2017).

One of the main difficulties is with modelling the turbulent heat fluxes. Typical Monin–Obukhov Stability Theory (MOST) assumes constant fluxes in the surface layer and so-called z-less scaling (Monin and Obukhov, 1954; Wyngaard and Coté, 1972) and its limits of applicability have been discussed (Grachev et al., 2008). This has led to the recognition of different turbulent regimes. The first, called the weakly stable regime, is fully consistent with MOST. In this regime, the turbulent heat fluxes increase with increasing temperature gradient because more heat is available to be transported. The inertial range in the turbulence spectra is well defined, and exhibits a Kolmogorov slope of -5/3 (Kaimal and Finnigan, 1994). The other is the strongly stable regime, where turbulent sensible heat fluxes instead decrease with increasing temperature gradient, because the





effect of strong stability lead to a turbulence decay. In this regime, Kolmogorov turbulence becomes intermittent and driven by processes at larger time scales such as the Coriolis force (Grachev et al., 2008) or gravity waves (Sorbjan and Czerwinska, 2013). However, it does not disappear entirely so that the flow never becomes laminar (Grachev et al., 2013).

There is general agreement on the nature of these two turbulence regimes (although sometimes, a third "transitional" regime
is considered). However, the separation between the two is debated: traditionally, the Richardson number ($R_i$) or Monin–Obukhov parameter ($\zeta$) are used. Grachev et al. (2013), for example, suggested that a gradient or flux Richardson number of $R_i = 0.2$ was a lower threshold for the strongly stable state, while Mahrt et al. (2014) found that $\zeta = 0.06$ separated the two states. More recent works have focused on the impact of wind speeds, or wind shear, on determining the regime (Sun et al., 2012; van de Wiel et al., 2007; van de Wiel et al., 2012), in a framework called Minimum Wind speed for Sustainable
Turbulence (MWST). For example, van Hooijdonk et al. (2015), building on the work of van de Wiel et al. (2012), used external forcings to the surface layer (such as a constant wind speed, replacing the synoptic pressure gradient, and downwards radiative fluxes) to determine a new parameter called the shear capacity. This parameter has been found to better predict the stability regime than the traditional local parameters such as $\zeta$ or $R_i$. In this new framework, the stability regime is not a feature solely of the turbulence, but of the surface layer as a whole.

Determining the stability regime and the turbulent heat fluxes is, however, only one part of determining the SBI strength. This depends on the surface temperature, which is in turn determined by the surface energy budget (SEB). Analysis of measurements in the Antarctic has shown that plotting $\Delta T$ (the temperature difference between the surface and 10 m) versus the wind speed at 10 m under clear-sky winter conditions reveals two distinct regimes, separated by a transition: one at low wind speeds and high $\Delta T$, and the other at high wind speeds and low $\Delta T$ (Vignon et al., 2017). This characteristic shape was termed 'S' shape
(although the 'S' is technically backwards) because the transition exhibited some non-monotonous behaviour. The transition between the two regimes was found to agree well with predictions from MWST. Drawing on these studies, a small analytical model was developed by van de Wiel et al. (2017) and shown to reproduce the 'S' shape.

MWST therefore offers a promising framework for the analysis and modelling of SBIs. For the moment, however, these analyses have been restricted to the extreme conditions of Antarctica, where the surface is vegetation-free snow and ice. The
Arctic and sub-Arctic also experience regular inversions with strong implications on pollution dispersion. However, a large part of this region is covered by forest, which is known to impact the turbulent heat fluxes (Batchvarova et al., 2001). For example, unstable stratification may remain within the canopy layer even when overlying air layer is very stable (Jacobs et al., 1992), and gradients directly above the canopy may be modified by the roughness sublayer (Mölder et al., 1999; Babić et al., 2016). Forest canopies also act as grey bodies, both emitting and absorbing longwave fluxes. In seeking to extend the use of MWST,
it is therefore important to consider the impact of trees. Another important question concerns the coherence of meso-scale models with MWST. Vignon et al. (2018), for example, showed that the LMDZ model reproduced an 'S' shape transition of surface temperature gradient with wind speed, with the shape of the transition depending on the stability function used. This represents a promising new framework for improving the representation of surface layer temperature inversions.

This paper therefore has two coupled aims. The first is to investigate the impact of wind speed on the temperature gradient
in clear-sky, winter conditions over a forest surface using a multiyear observational dataset from a forest of Interior Alaska.





The second is to evaluate and improve the performance of WRF schemes in representing the temperature gradient at this forest site. Here, the parts of the WRF code responsible for calculating the surface temperature gradient are referred to as the Surface Energy Budget and Surface Layer (SEB-SL) model, because they calculate the turbulent energy exchanges in the surface layer and solve the surface energy budget. In WRF, the SEB-SL model is often split into two parts: the surface layer scheme and the land surface model. In the Sect. 2, conceptual SEB-SL models will be introduced and used to shed light on two different WRF SEB-SL schemes. In Sect. 3, the measurements from the Ameriflux Poker Flats Research Range near Fairbanks, Alaska are presented. Modifications are proposed to the two WRF SEB-SL schemes, and the methodology for evaluating the original and modified schemes is explained. Lastly, results are presented (Sect. 4).

## 2 Presentation of SEB-SL models

In this section, conceptual models of the surface layer are presented (Sect. 2.1). These models are used to gain insight on the impact of different variables (and especially, of the wind speed) on the resulting surface–air temperature gradient. In Sect. 2.2, two WRF SEB-SL models, and their divergences from the conceptual models, are presented.

### 2.1 Conceptual model

#### 2.1.1 Model presentation

van de Wiel et al. (2017) developed a single layer conceptual SEB-SL model to study the impact of the wind speed on the near-surface temperature gradient. In the presence of trees or other tall vegetation, however, the introduction of a second layer becomes necessary. Here, such a model is developed. It is composed of the surface, a "canopy" layer where the air is in thermodynamic equilibrium with the vegetation, and an overlying air layer (Fig. 1). The effect of the canopy on the longwave radiative and turbulent fluxes are then taken into account. In the following, the equations and notations draw on the one layer model of van de Wiel et al. (2017). The surface emissivity is assumed to be equal to 1, which is a good approximation for snow covered surfaces, and the shortwave radiation is neglected.

The surface energy balance equation in this system can be written as (Appendix A):

$$
\begin{aligned}
-(1-\epsilon_c)Q_i - \Lambda_s(T_a - T_g) + \Delta T_{cs}\left[\rho C_p C_{D,c} U_c + 4\sigma T_a^3 + \Lambda_s\right] & \\
+ \Delta T_{ac}\left[(1-\epsilon_c)4\sigma T_a^3 + \Lambda_s\right] & \\
= 0 &
\end{aligned}
\tag{1}
$$

where $\Delta T_{cs} = T_c - T_s$ is the difference between the surface temperature ($T_s$) and the air temperature at canopy height ($T_c$). $\Delta T_{ac} = T_a - T_c$ is difference between the canopy temperature and the air temperature at height $z_a$, corresponding to the top of the surface layer ($T_a$). $T_g$ is the ground temperature (Fig. 1). $Q_i = -\text{LW}_d + \sigma T_a^4$ (with $\text{LW}_d$ the downwards longwave flux) is termed the isothermal net radiation: indeed, it is equal to the net longwave flux if $T_s = T_a$ (Holtslag and Bruin, 1988). $\Lambda_s = \frac{\lambda_s}{d_s}$, where $\lambda_s$ is the snow conductivity and $d_s$ the snow depth. $\rho$ is the air density, $C_p$ the heat capacity of the air, $U_c$ the wind speed





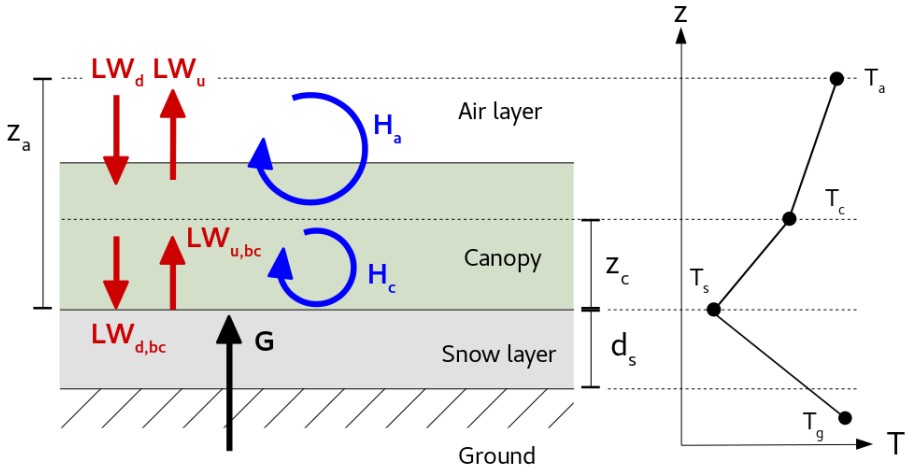

**Figure 1.** Schematic of the two layer model described in this section. $\text{LW}_d$ and $\text{LW}_u$ are the downwards and upwards longwave fluxes above the canopy, and $\text{LW}_{d,bc}$ and $\text{LW}_{u,bc}$ are the downwards and upwards longwave fluxes below the canopy. $\text{H}_a$ is the turbulent sensible heat flux between the canopy and overlying air. $\text{H}_c$ is the turbulent sensible heat flux between canopy and surface. G is the conduction flux through the snow.

at height $z_c$. For the time being, $U_c$ will be very roughly estimated to be proportional to $U_a$: for example, $U_c = 0.25 \cdot U_a$. $\epsilon_c$ is the canopy emissivity and $C_{D,c}$ is the turbulent diffusion coefficient for the canopy to surface heat exchange.

A second equation can be obtained by considering the canopy energy balance (Appendix A):

$$
\begin{aligned}
-Q_i - \Lambda_s(T_a - T_g) + \Delta T_{cs}\left[(1-\epsilon_c)4\sigma T_a^3 + \Lambda_s\right] \\
+ \Delta T_{ac}\left[4\sigma T_a^3 + \Lambda_s + \rho C_p C_{D,a} U_a\right] \\
= 0
\end{aligned}
\tag{2}
$$

with $U_a$ the wind speed at height $z_a$ and $C_{D,a}$ is the turbulent diffusion coefficient for the air–canopy heat exchange. This is very similar to the one layer model of van de Wiel et al. (2012), except that the energy source term (here, the first line of Eq. 2) has an additional term, which is proportional to $\Delta T_{cs}$. The difficulty in solving Eq. 1 and 2 to obtain $\Delta T_{cs}$ and $\Delta T_{ac}$ is that the turbulent diffusion coefficients depend on the stability:

$$
\begin{aligned}
C_{D,c} &= \frac{\kappa^2}{[\log(\frac{z_a}{z_{0s}}) - \psi(\zeta) + \psi(\frac{z_{0s}}{L})]^2} \\
C_{D,a} &= \frac{\kappa^2}{[\log(\frac{z_a-d}{z_{0c}}) - \psi(\zeta) + \psi(\frac{z_{0c}}{L})]^2}
\end{aligned}
\tag{3}
$$

with $\kappa = 0.4$ the van Kármán constant, and $z_{0s}$ and $z_{0c}$ the roughness lengths of snow or of the canopy respectively. $d$ is the displacement height due to the presence of the canopy. Here, the snow and canopy momentum roughness lengths $z_{0m,s/c}$



| Physical constant | Name | Value | Unit |
|---|---|---|---|
| $\sigma$ | Stefan-Boltzmann constant | $5.67\ 10^{-8}$ | $\text{W K}^{-4}$ |
| **Parameter** | **Name** | **Value** | **Unit** |
| $\rho$ | Air density | 1.2 | $\text{kg m}^{-3}$ |
| $\lambda_s$ | Snow heat conductivity | 0.3 | $\text{W m}^{-1}\ \text{K}^{-1}$ |
| $C_p$ | Heat capacity of air | 1005 | $\text{J K}^{-1}\ \text{kg}^{-1}$ |
| $z_{0s}$ | Snow roughness length | 0.002 | m |
| **Inputs** | **Name** | **Typical value range** | **Unit** |
| $\epsilon_c$ | Canopy emissivity | $0-1$ | - |
| $U_a$ | Wind speed at $z_a$ | $0-15$ | $\text{m s}^{-1}$ |
| $T_a$ | Air temperature at $z_a$ | $243.15-273.15$ | K |
| $Q_i$ | Isothermal net radiation | $20-80$ | $\text{W m}^{-2}$ |
| $d_s$ | Snow depth | $0.1-1$ | m |
| $T_g$ | Ground temperature | $263.15-273.15$ | K |
| $U_c$ | Canopy wind speed | $<U_a$ | $\text{m s}^{-1}$ |
| $z_{0c}$ | Canopy roughness length | $0.3-1.$ | m |
| **Outputs** | **Name** | | **Unit** |
| $\Delta T_{cs}$ | Canopy-surface temperature difference | | K |
| $\Delta T_{ac}$ | Air-canopy temperature difference | | K |
| $\Delta T_{as}$ | Air-surface temperature difference | | K |
| $C_{D,a}$ | Turbulent diffusion coefficient (air–canopy) | | - |
| $C_{D,c}$ | Turbulent diffusion coefficient (canopy–surface) | | - |
| $L$ | Monin–Obukhov length | | m |

**Table 1.** List of the constants, parameters and variables (both input and output) used in the conceptual model. For the inputs, a typical range of values for the Arctic winter (in clear-sky conditions) is indicated.

are assumed to be equal to the heat roughness length $z_{0h,s/c}$, both referred to as $z_{0,s/c}$. $\zeta = z/L$ is the Obukhov parameter with $L$ the Monin–Obukhov length. $\psi$ is the integral stability function, which tends to 0 when $\zeta \approx 0$, and tends to infinity with increasing $\zeta$. The turbulent diffusion coefficients therefore tend to $\kappa^2/\log(\frac{z}{z_0})^2$ at weak stability and 0 at strong stability. Many different expressions of $\psi$ are found in the literature (Businger et al., 1971; Holtslag and Bruin, 1988). Usually, these are classified as "short-tail" (i.e., with a very sharp increase/decrease so that $C_D$ quickly drops to 0 at increasing stability) or

"long-tail" (i.e., the transition is smoother so that some turbulent sensible heat flux is maintained for longer). There are also other ways to estimate the below canopy turbulent diffusion coefficient $C_{D,c}$, for example by assuming an exponential wind profile in the canopy as in Mahat et al. (2013). However, Eq. 3 is the simplest expression and will serve for illustrative purposes.





### 2.1.2 Weakly and strongly stable limits

First insights into the behaviour of $\Delta T_{cs}$ and $\Delta T_{ac}$ can be gained by studying the asymptotic cases: the weakly and strongly
stable limits. Here only the case where $\epsilon_c = 1$ (corresponding to an opaque canopy) is considered. In this situation, and if
turbulence is completely collapsed (i.e. $C_{D,a} = C_{D,c} = 0$), Eqs. 1 and 2 lead to the following values for the temperature
gradients:

$$\Delta T_{ac} = \frac{Q_i \left[ 1 + 1/(4\sigma T_a^3) \right] + \Lambda_s (T_a - T_g)}{4\sigma T_a^3 + 2\Lambda_s}$$
$$\Delta T_{cs} = \frac{-Q_i/(4\sigma T_a^3) + \Lambda_s (T_a - T_g)}{4\sigma T_a^3 + 2\Lambda_s}$$

(4)

and therefore,

$$\Delta T_{as} = \Delta T_{ac} + \Delta T_{cs} = \frac{Q_i + 2\Lambda_s (T_a - T_g)}{4\sigma T_a^3 + 2\Lambda_s}$$    (5)

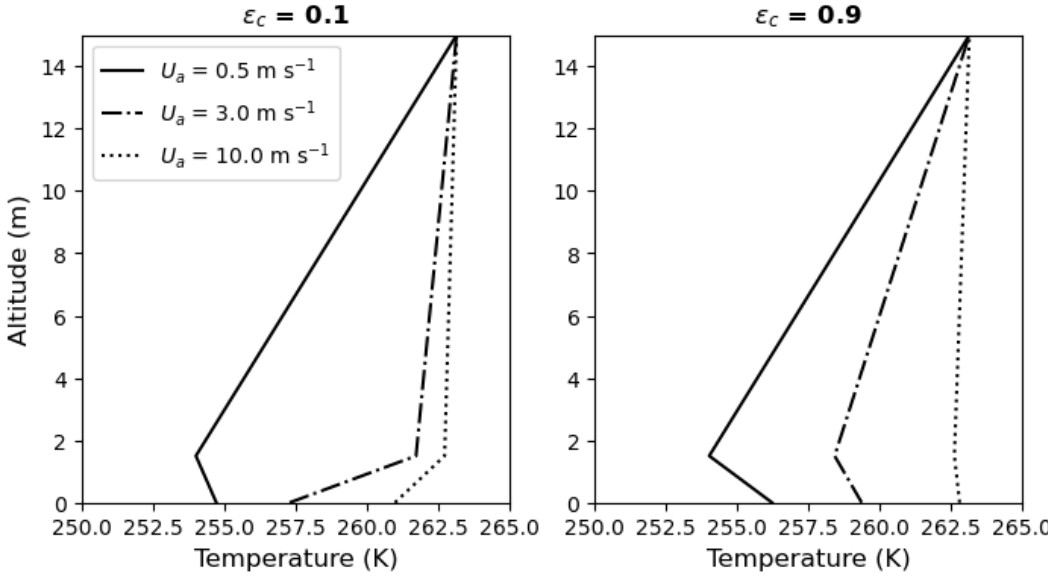

**Figure 2.** Profile calculated by the theoretical model for $Q_i = 50$ W m$^{-2}$, $T_a$ = -10 °C, $T_g$ = -2 °C and $\Lambda_s = 1$ W m$^{-2}$ K$^{-1}$, for three values
of the wind speed $U_a$ (0.5, 3 and 10 m s$^{-1}$). The turbulence was solved iteratively using the Ameriflux stability function. The only difference
between the two graphs is the canopy emissivity: left, $\epsilon_c = 0.1$ and right, $\epsilon_c = 0.9$.

The total temperature gradient $\Delta T_{as}$ (between air and surface) in Eq. 5 is very similar to the one layer case, but with an
"equivalent snow conductivity" twice the real value.

It can also be noted that $\Delta T_{ac}$ will usually be positive for typical Arctic winter values of the different parameters (Table 1)
because radiation is the dominant process, unless the snow cover is very thin. For example, in a very cold, high synoptic





pressure situation with $Q_i = 70$ W m$^{-2}$, $T_a$ = -20°C and $T_g = 0$°C, $\Delta T_{ac}$ will be positive unless the snow depth is less than 7 cm. Similarly, in a warmer, cloudier situation with $Q_i = 30$ W m$^{-2}$, $T_a$ = -10°C and $T_g = 0$°C, $\Delta T_{ac}$ will be positive unless the snow depth is less than 8 cm. For the same reason, $\Delta T_{cs}$ will usually be negative. In short, the very stable case is characterised by a temperature decrease from surface to canopy, and an increase from canopy to the overlying air. This is illustrated in Fig. 2 (continuous line, corresponding to $U_a = 0.5$ m s$^{-1}$).

This is contradictory with the idea that $C_{D,c}$ collapses to 0, because in the presence of a negative temperature gradient buoyancy effects may generate turbulence without significant mechanical shear. Indeed, solving Eqs. 1 and 2 numerically for $U_c = U_a = 0.001$ m s$^{-1}$ (using appropriate schemes for calculating the turbulent diffusion coefficients such as described in Sect. 2.2.1) shows that $C_{D,c}U_c$ maintains a value of around 0.0017 m s$^{-1}$. Therefore, while the surface layer as a whole may be considered strongly stable (because $T_a - T_s$ is very large), this may not be the case for the canopy layer. This is in agreement

with Batchvarova et al. (2001), which found that the canopy layer may remain unstable even when the air aloft is very stably stratified.

### 2.1.3 Transition between the weakly and strongly stable limits

Next, the turbulence is solved iteratively using the Ameriflux stability function (Sect. 3.1) over a complete range of $U_a$ values. This makes it possible study the behaviour of the inversion outside of the weakly and strongly stable regimes and for different

values of the canopy emissivity. The result of this estimation is shown in Fig. 3a,b for three values of $\epsilon_c$. $\Delta T_{ac}$ exhibits the same 'S' shape as in the one layer conceptual model of van de Wiel et al. (2017). On the other hand, $\Delta T_{cs}$ is larger in the weakly stable regime than in the strongly stable regime (where it is negative, in coherence with the above discussion), and its shape is more dependent on values of the canopy emissivity. For $\epsilon_c = 1$, $\Delta T_{cs}$ appears to tend to zero at large values of $U_a$ while keeping negative values. On the other hand, for $\epsilon_c = 0.5$ and 0.1, it turns positive before decreasing with increasing

wind speeds, therefore reaching a maximum somewhere between 2 and 6 m s$^{-1}$. In sum, the total temperature difference ($\Delta T_{as}$) decreases more slowly than in the one-layer model, leading to a less marked 'S' shape, even when accounting for the long-tailed function chosen.

This behaviour can be understood in the following way. At low wind speeds, the dominant process influencing the canopy layer is radiation: it emits more than it receives, and therefore loses its heat both to the surface and the air above. As a result,

it is colder than both. Although some turbulence remains due to buoyancy, this is not enough to compensate the radiative heat loss. At high wind speeds, on the other hand, the whole SL is well mixed. Turbulence is the dominant process, linking the canopy layer to both the surface and the air above and maintaining their temperatures close.

Starting from the strongly stable state, a small increase in wind speed will lead to increased turbulence mixing and a positive heat flow from the air above to the canopy. The canopy temperature will therefore increase. If the canopy has a strong emissivity,

this increase in temperature will lead to increased radiation downwards to the surface and a corresponding increase in surface temperature. Canopy and surface will therefore warm at relatively the same pace (Fig. 2, right). On the other hand, if the emissivity of the canopy is low, it will not as easily convert its increased temperature into radiation. Its temperature will therefore increase rapidly without contributing to warming the surface, leading to a high $\Delta T_{cs}$ (Fig. 2, left). As the wind speed





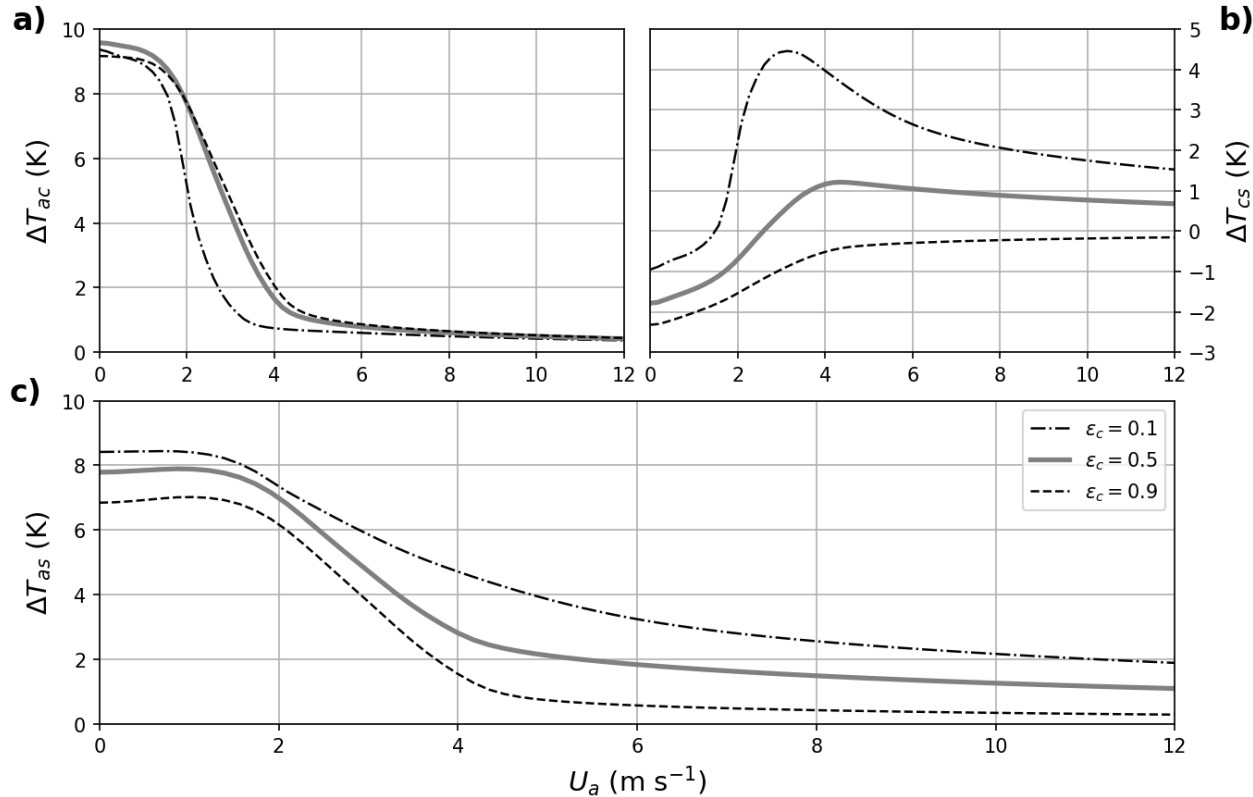

**Figure 3.** Panel a: $\Delta T_{ac}$ as a function $U_a$, as calculated by the conceptual model. The different line styles correspond to different values of the canopy emissivity. The transition between the two regimes has been calculated using the MYJ algorithm with the Ameriflux stability function (Sect. 2.2.1). All curves have been calculated using $Q_i = 50$ W m$^{-2}$, $T_a = $ -10 °C, $T_g = $ -2 °C and $\Lambda_s = 1$ W m$^{-2}$ K$^{-1}$. Panel b: same, for $\Delta T_{cs}$. Panel c: same, for $\Delta T_{as}$.

continues to increase, the canopy temperature will eventually be more or less equal to the temperature of the air above, and the
surface continues to warm, thus leading to a decrease in $\Delta T_{cs}$.

## 2.2 WRF SEB-SL modules

Within the WRF code, the SEB-SL model is split into two parts. First, the surface layer module calculates the turbulent diffusion coefficient. Then the land-surface model uses the turbulent diffusion coefficient to solve the SEB and determine the surface temperature. Many different schemes are available for each module. In this work, we have chosen to focus on the Noah Land
Surface Model (Noah-LSM) (Chen and Dudhia, 2001; Ek et al., 2003), used in conjunction with the Mellor-Janjić-Yamada (MYJ) surface layer scheme (Janjić, 1994); and on the Noah MultiPhysics scheme (Niu et al., 2011; He et al., 2023), which is nominally a land surface model but actually functions as an entire SEB-SL model, as it calculates its own turbulent diffusion coefficients. In the rest of this article, the Noah-LSM and MYJ combination will be termed 'Noah-MYJ'.





### 2.2.1 Noah-MYJ

When there is a snow cover, Noah-LSM functions very similarly to the one layer model of van de Wiel et al. (2017). The snowpack is considered as a single layer, while the soil is subdivided into four layers for which the heat diffusion equation is solved, yielding the topmost soil layer temperature $T_g$. In the MYJ scheme, $C_D$ is calculated as:

$$C_D = \frac{\kappa^2}{(\log(\frac{z}{z_{0m}}) - \psi(\zeta) + \psi(\frac{z_{0m}}{L})) \cdot (\log(\frac{z}{z_{0h}}) - \psi(\zeta) + \psi(\frac{z_{0h}}{L}))}$$

$$z_{0h} = z_{0m} e^{-\kappa \cdot 0.1 \cdot (1 + \frac{R_b}{R_c}^2) \sqrt{u^* z_{0m}/\nu_a}} \tag{6}$$

with $\nu_a = 1.47 \times 10^{-5}$ m$^2$ s$^{-1}$ the air kinematic viscosity, $R_b$ the bulk Richardson number and $R_c$ a critical Richardson number, here equal to 0.505. $u^*$ is the friction velocity. The momentum roughness length $z_{0m}$ is fixed according to the land-use type and vegetation while heat roughness length $z_{0h}$ depends on the stability (through the Richardson number). Note that because this is a one layer model, there are no separate snow and canopy roughness lengths.

$\psi$ has the following expression in stable conditions (i.e. $\zeta \geq 0$):

$$\psi(\zeta) = 0.7 \cdot \zeta + 0.75 \cdot \zeta \cdot (6 - 0.35\zeta) \cdot e^{-0.35\zeta} \tag{7}$$

This is similar, but not equal, to the Holtslag integral stability function (Holtslag and Bruin, 1988) up to values of $\zeta \approx 1$. Indeed, $\zeta$ has a set maximum value of 1. This means that $C_D$ never goes to 0, some turbulent sensible heat flux is always maintained, and the very stable regime is not independent of $U_a$. For low values of the momentum roughness length the distinction between very and weakly stable regimes even completely disappears, with $\Delta T_{as}$ instead decreasing almost linearly as a function of $U_a$. Solving this form of the equation for $C_D$ requires that $L$ to be known, which in turn requires knowledge of both $u^*$ and the turbulent sensible heat flux and thus $C_D$. The solving procedure is therefore iterative.

### 2.2.2 Noah-MP

Recently, Noah-MP was introduced as an updated version of Noah-LSM introducing, among others, a vegetation energy balance, a layered snowpack, and soil moisture – groundwater interaction (Niu et al., 2011; He et al., 2023). Each grid node is divided into a vegetated and a non-vegetated fraction. The non-vegetated fraction surface temperature is calculated similarly to Noah-LSM, except that the snowpack is divided into up to three different layers and the ground heat flux is calculated through the topmost snow layer only. The vegetated fraction calculation is a more complex version of a 2-layer model, where the vegetation temperature is considered to be different from the air temperature in the canopy. The vegetation acts as a grey body with emissivity $\epsilon_v$ and exchanges sensible heat with the canopy air. The canopy air is transparent to longwave radiation, simply exchanging sensible heat with the surface, the overlying air and the vegetation. In short, the radiative and sensible heat budgets of the canopy layer in Sect. 2.1 are separated. In practice, however, the temperature difference between the vegetation and the canopy air did not exceed 0.5 K during our runs, so that a simple 2-layer model provides a good approximation for the behaviour of Noah-MP. Therefore, we do not not detail the calculation of the tree – canopy air sensible heat exchange.





| | Evergreen Needle-leaf Forest | Mixed Forest | Wooded Tundra | Mixed Tundra | Ameriflux PRR site |
|---|---|---|---|---|---|
| LAI ($m^2\ m^{-2}$) | 4.4 | 2.4 | 1.2 | 0.7 | 0.73 |
| $h_{can}$ (m) | 20 | 16 | 4 | 2 | 3 |
| $z_{0m,c}$ (m) | 1.09 | 0.8 | 0.3 | 0.2 | 0.4 |
| $d$ (m) | 13 | 10.4 | 2.6 | 1.3 | 1.4 |
| $f_{veg}$ | 0.7(table) 0.9(LAI) | 0.8(table) 0.7(LAI) | 0.6(table) 0.45(LAI) | 0.6(table) 0.3(LAI) | $f_{veg}\epsilon_v \approx 0.15$ |
| $\epsilon_v$ | 0.99 | 0.91 | 0.7 | 0.5 | |

**Table 2.** Noah-MP surface characteristics for four different land use types: Evergreen Needleleaf Forest, Mixed Forest, Wooded Tundra and Mixed Tundra. The characteristics include the Leaf Area Index (LAI), canopy height ($h_{can}$), canopy momentum roughness length ($z_{0m,c}$), displacement height $d$, vegetation fraction ($f_{veg}$) and vegetation emissivity ($\epsilon_v$). The last column shows the surface characteristics at the Ameriflux Poker Flats Research Range in Interior Alaska, which will be presented in Sect. 3.1.

The turbulent diffusion coefficient for the canopy to overlying air sensible heat exchange, $C_{D,a}$, is calculated using the "original Noah" scheme with a roughness length and displacement height which depend on the land use category; the displacement height is calculated as $d = 0.65 \cdot h_{can}$, where $h_{can}$ is the canopy top height. This "original Noah" scheme is identical to the MYJ scheme described above, except that it uses the Businger-Dyer stability function (Businger et al., 1971). $C_{D,c}$ is calculated by assuming an exponential wind profile, similar to what is described in Mahat et al. (2013):

$$K_h = \frac{\kappa^2 \cdot U_c \cdot (h_{can} - d)}{\log((z_a - d)/z_{0m,s})}$$
$$C_{D,c}U_c = \frac{K_h \cdot n}{h_{can} \cdot e^n \left( \exp\left[-nz_c/h_{can}\right] - \exp\left[-n(d + z_{0m,c})/h_{can}\right] \right)} \tag{8}$$

where $n$ is the exponential decay coefficient, which depends on the leaf area index (LAI), canopy top height, and stability, and $z_{0m,s}$ is the below-canopy ground roughness length. In this article, the below canopy ground cover is always assumed to be snow.

The total grid box surface temperature is then calculated from the two values obtained for the vegetated ($T_{s,v}$) and non-vegetated parts ($T_{s,nv}$):

$$T_s = f_{veg}T_{s,v} + (1 - f_{veg})T_{s,nv} \tag{9}$$

where $f_{veg}$ is the vegetation fraction in each model grid box. There are multiple calculation options for this parameter in Noah-MP. It can either be taken from the vegetation parameter table, which is also used by Noah-LSM, or it can be determined from the LAI using the following formula:

$$f_{veg} = 1 - e^{-0.52LAI} \tag{10}$$

LAI itself can either be taken from the Noah-MP parameter table or determined "dynamically" by a carbon budget subroutine. Typical values of LAI and $f_{veg}$ for different land-use categories are shown in Table 2.




It should be noted that the vegetation emissivity used in Noah-MP is not equivalent to the canopy emissivity in the simple 2-layer model described in Sect. 2.1. In effect, Noah-MP supposes that the vegetated fraction has an emissivity of $\epsilon_v$ and the non-vegetated fraction has an emissivity of 0: the average canopy emissivity (such as is used by the model in Sect. 2.1) is therefore $\epsilon_c = f_{veg}\epsilon_v$.

## 3  Methodology

First, long-term measurements from the Ameriflux Poker Flats Research Range (PRR) site are introduced (Sect. 3.1). Then the modifications made to the WRF SEB-SL schemes are presented and the evaluation method for the schemes is explained (Sect. 3.2).

### 3.1  Measurements at the Ameriflux Poker Flats Research Range

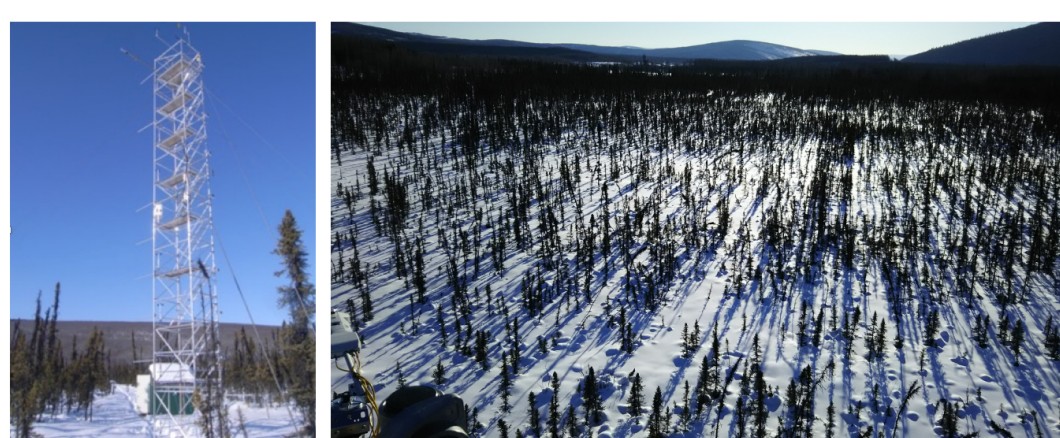

**Figure 4.** Left: photo of the 17 m high measurement tower at the Poker Flats Research Range Ameriflux site. Right: photo of the PRR site as seen from the measurement tower. Credit: Lisa Johnson.

The Ameriflux PRR site is located in the Poker Flats Research Range (65°07'24.4" N, 147°29'15.2" W), around 30 km away from Fairbanks (Interior Alaska). It has been operating since 2010, when it was established as part of the JAMSTEC-IARC Collaboration Study (JICS) (Sugiura et al., 2011) and its data is made available online on the Ameriflux website (Kobayashi et al., 2019) (https://ameriflux.lbl.gov/sites/siteinfo/US-Prr). Its 17 m measurement tower is implanted in a black spruce forest with sparsely distributed and short trees (Fig. 4). The tree density, as measured in 2010, was 3967 trees ha$^{-1}$ and the average tree height was 2.44 m: the tallest tree was 6.4 m but 75% of trees were shorter than 3 m (Nakai et al., 2013). The LAI was 0.73 (Nakai et al., 2013). Both of these values are much smaller than those found in Noah-MP for evergreen forests (Table 2).

The Ameriflux PRR measurements which will be used here are summarised in Table 3. These include wind speeds and temperatures at 8 different heights (from 1.5 to 16 m), as well as turbulent and radiative fluxes. The surface temperature was calculated from the radiative flux measurements at 1.3 m, assuming a snow surface emissivity of 0.99. As this study focuses





| Variable | Instrument | Measurement altitude (m) |
|---|---|---|
| Wind speed (m s$^{-1}$) | 010C (MetOne, USA) | 1.5, 3, 4.5, 6, 7.5, 9, 13, 16 |
| Temperature (K) | HMP155 (Vaisala, Finland) | 1.5, 3, 4.5, 6, 9, 11, 13, 16 |
| Turbulent sens. heat flux (W m$^{-2}$) | WindMaster Pro (Gill, UK) | 1.9, 11 |
| Friction velocity (m s$^{-1}$) | WindMaster Pro (Gill, UK) | 1.9, 11 |
| Radiative fluxes (W m$^{-2}$) | CNR4 (Kipp & Zonen, Netherlands) | 1.3, 16 |
| Snow depth (m) | SR50A (Campbell Sci., USA) | 0 |
| Soil temperature (K) | 107 (Campbell Sci., USA) | -0.05, -0.1 |

**Table 3.** Meteorological variables measured at the Ameriflux PRR site, including instruments and measurement heights (Nakai et al., 2013). Note that wind speed is also measured at 11 m, with the sonic anemometer. Nakai et al. (2013) indicates that temperature was also measured at 7.5 m, but these measurements do not appear to be available on the Ameriflux website.

on the clear-sky surface layer in wintertime conditions, the data was curated accordingly. Only time points in the months of November – March with snow depth greater than 10 cm were kept. As no measurements of the cloud cover are available at the PRR site, clear-sky instants were defined as those with net longwave radiation less than -30 W m$^{-2}$. Indeed, as is typical in high-latitude site, the net longwave flux ($R_n$) distribution at the PRR site was bimodal; the low $R_n$ mode was considered to correspond to the absence of clouds and the high $R_n$ mode to their presence. As the PRR site is located slightly below the Arctic circle, there is still some solar radiation at the surface in the winter time. In order to simplify the analysis, only time points with downwelling shortwave radiation less than 30 W m$^{-2}$ were kept; as the snow albedo is very high, this corresponds to a net shortwave flux less than 5 W m$^{-2}$ and therefore to negligible shortwave impact. Lastly, measurements with latent heat flux greater than 5 W m$^{-2}$ in absolute value were discarded.

The average emissivity of the canopy layer $\epsilon_c = f_{veg}\epsilon_v$ can be calculated from above and below canopy radiation measurements:

$$\mathrm{LW}_{d,bc} - \mathrm{LW}_u = (1 - f_{veg}\epsilon_v)\left[\mathrm{LW}_d - \mathrm{LW}_{u,bc}\right] \tag{11}$$

As shown in Fig. 5c, this gives a best estimation of $\epsilon_c \approx 0.15$. This is coherent with the Noah-MP calculation of $f_{veg}$ and $\epsilon_v$ as a function of LAI. Indeed, measured LAI at the PRR site is 0.73, yielding $f_{veg} \approx 0.3$ (Eq. 10) and $\epsilon_v \approx 0.5$.

The value of $z_{0m,c}$ can also be calculated from the sonic anemometer data. Indeed, at weak stability ($\zeta \ll 1$), the wind profile is approximately logarithmic:

$$U_a \approx \frac{u^*}{\kappa} \log\left(\frac{z - d}{z_{0m,c}}\right)$$

Therefore, $d$ and $z_{0m,c}$ can be determined through a linear regression of $e^{\kappa U_a/u^*}$ against $z$ when the data is restricted to values of $\zeta < 10^{-2}$. Here, $z_{0m,c}$ was found to be 0.39 m with $d = 1.4 \pm 1.4$ m (Fig. 5b), which makes sense for a forest environment with short trees. The integral stability function $\psi$ can also be determined from the data (assuming, as is often done, that it is the same for momentum and heat). In order to do this,

$$\Psi = -\psi(\zeta) + \psi(\frac{z_{0m,c}}{L}) = \frac{U_a}{u^*}\kappa - \log\left(\frac{z - d}{z_{0m,c}}\right)$$




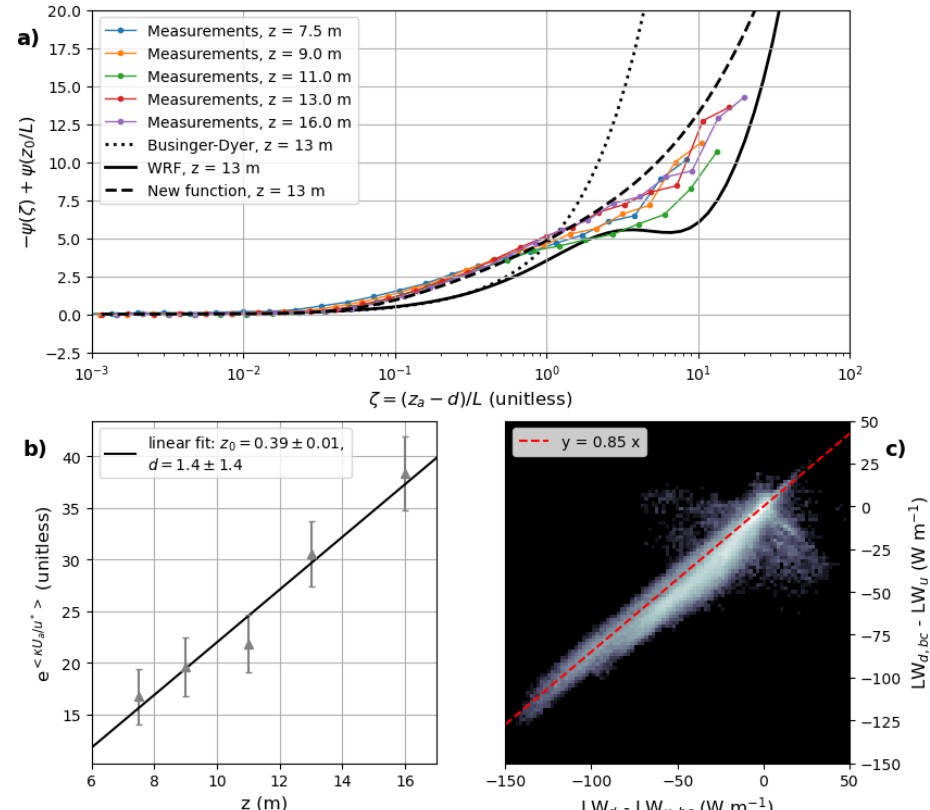

**Figure 5.** Panel a: momentum integral stability function as a function of $\zeta$ determined from the PRR site measurement (coloured lines) and calculated using the Businger-Dyer and WRF formulations (black lines). The dashed line corresponds to the new determined function for $\psi$ (Eq. 12). Panel b: determination of the PRR site above-canopy momentum roughness length and displacement height. Panel c: 2D histogram of $LW_{d,bc} - LW_u$ vs $LW_d - LW_{u,bc}$. The red line corresponds to $y = 0.85x$, yielding a canopy emissivity of 0.15.

is plotted as a function of $\zeta$ (Fig. 5a). Because $z_{0m,c}/L$ is negligible compared to $\zeta$, $\Psi$ is approximately equal to $-\psi(\zeta)$ at the

first order. Here, we found that measured $\psi$ is more gradual (i.e., more long-tailed) than the Businger-Dyer (Businger et al., 1971) or WRF (Eq. 7) functions. In fact, when plotted on a log-log scale, it became apparent that $\Psi$ was proportional to $\sqrt{\zeta}$ at low values of $\zeta$, and proportional to $\zeta^2$ at high values of $\zeta$. This differs from the often used z-less scaling, which implies that $\psi$ must be proportional to $\zeta$ at least up to $\zeta \approx 0.1$ (Monin and Obukhov, 1954; Grachev et al., 2013) as is the case for the WRF and Businger-Dyer functions. Measurement error may explain some of the difference at $\zeta < 0.1$ as $\Psi$ is very small in these

conditions.

For our purposes, we determined an analytical expression of $\psi$ to best fit to the $\Psi$ measurements and left aside the question of the z-less scaling. The following expression for $\psi$ was therefore considered:





$$\psi(\zeta) = -a \cdot \zeta^{r(\zeta)}$$
$$r(\zeta) = 0.75 \cdot \arctan(b \cdot \zeta - c)\frac{2}{\pi} + 1.25 \tag{12}$$

This was chosen because $\arctan(x)$ tends to $\pm\frac{\pi}{2}$ when $x$ tends to $\pm\infty$, with a smooth transition around $x = 0$. $r(\zeta)$ will

therefore tends to 0.5 at low values of $\zeta$ and 2 at high values of $\zeta$, similar to observations. The $b$ and $c$ coefficients must be chosen so that the timing and speed of the transition between the $\sqrt{\zeta}$ and $\zeta^2$ asymptotes matches the observations. Values of $a = 5$, $b = 20$ and $c = 0.1$ were found to give a good fit to the observations (Fig. 5a). This expression of $\psi$ was termed the PRR stability function.

The Ameriflux PRR site characteristics are summarized in Table 2. Although the location of the PRR site is given to be

Evergreen Needleleaf Forest by the MODIS (MODerate resolution Imaging Spectroradiometer) land-use categories as well as by the Ameriflux website, its canopy height and turbulent and radiative characteristics are actually most similar to a Wooded or Mixed Tundra.

## 3.2 Modifications to the WRF SEB-SL schemes

| Short name | Model | Type | Parameters | Turbulent diffusion coefficient | Comments |
|---|---|---|---|---|---|
| **oMYJ** | Original MYJ + Noah-LSM | 1-layer | – | WRF stability function | Max $\zeta$ set to 1 |
| **mMYJ** | Modified MYJ + Noah-LSM | 1-layer | $d$ | PRR stability function | Max $\zeta$ set to 100 |
| **oMP** | Original Noah-MP | 2-layer | $d, z_{0m,c}, LAI$ | Businger-Dyer stability function (top layer); Eq. 8 formulation (bottom layer) | Max $\zeta$ set to 1 – different vegetation and canopy air temp. |
| **mMP** | Modified Noah-MP | 2-layer | $d, z_{0m,c}, LAI$ | PRR stability function (top and bottom layer) | Max $\zeta$ set to 100 – $z_{0h,c} = 0.01 \cdot z_{0m,c}$ – $K = 5\times 10^{-4}$ m s$^{-1}$ – same vegetation and canopy air temp. |

**Table 4.** Summary of the four surface models evaluated in this study. For all four models, $z_{0m,c}$ was set to 0.4 m and $\lambda_s$ to 0.3 W m$^{-1}$ K$^{-1}$.

In this paper, two 1-layer and two 2-layer models are compared (Table 4). The 1-layer models include the original Noah-

MYJ (oMYJ), presented in Sect. 2.2, and a modified version of this scheme (mMYJ). The 2-layer models include the original Noah-MP (oMP) and a modified version of Noah-MP (mMP).





The guiding principle for the modifications to both original models was to improve the modelled dependency of the temperature inversion on the wind speed ('S' shape). This included removing the imposed maximum on $\zeta$, so that a truly stable regime is allowed to develop. The stability function was also modified to a more long-tail formulation (Sect. 3.1): this makes
the transition more gradual, and avoids the non-monotonicity associated to Eq. 7 at $\zeta > 1$. Furthermore, a displacement height is added in the mMYJ model.

Modifications implemented in mMP included forcing the vegetation and canopy air temperature to be equal, so that the energy balance for the vegetated part is as described in Sect. 2.1. The canopy to ground turbulent diffusion coefficient was also calculated as in the MYJ surface layer instead of using Eq. 8. Lastly, a constant coefficient $K = 5 \times 10^{-4}$ m s$^{-1}$ was added to
$C_{D,a}U_a$. This effectively imposes a lower limit on the turbulent diffusion coefficient in a gradual way, without having to force a minimum which would create a discontinuity. At wind speeds greater than 3 m s$^{-1}$, this constant coefficient is negligible compared to the calculated value of $C_{D,a}U_a$. It should be noted that in effect, the original Noah-MP also imposes such a limit through indirect methods (for example, by imposing a minimum value of 1 m s$^{-1}$ for $U_a$ or a maximum value of 1 for $\zeta$). The reasons for imposing a lower limit on the turbulence are explored further in Sect. 4.1.
In order to evaluate the models "offline", the oMYJ and oMP models were extracted from the WRF framework and recoded in Python in a minimal form, i.e. only the parts relating to the surface temperature calculation were kept. In particular, all latent heat flux calculations were ignored; the snow conductivity was assumed to be constant, equal to 0.3 W m$^{-1}$ K$^{-1}$; and snow depth and ground temperature were used as input variables rather than being calculated. The recoding was checked to be correct by comparing the calculated turbulent diffusion coefficients to the output of actual WRF runs for different wind speeds.
mMYJ and mMP were similarly coded in Python.

Input parameters to all four models are set to correspond to the characteristics of the PRR site as determined in Sect. 3.1: i.e., $d = 1.4$ m, $z_{0m,c} = 0.4$ m and $z_{0m,s} = 0.002$ m. For oMP and mMP, f$_{veg}$ was determined from the LAI using Eq. 10. For the two modified versions, the stability function used is the one determined from the Poker Flats Research Range measurements (Sect. 3.1).

Here, the top of the surface layer was considered to be the top of the measurement tower, i.e. $z_a = 16$ m. The five input variables to the Python models are the measured air temperature at 16 m, wind speed at 16 m, downwards longwave flux above the canopy, snow depth, and ground temperature. The output is the surface temperature (and canopy temperature for the 2-layer models). Running the models over the entirety of the curated PRR dataset yielded 5412 modelled values of surface temperature, which were then compared to the corresponding 5412 measured values of $T_s$. The results are analysed in Sect. 4.2.

## 4   Results

### 4.1   Link between temperature gradients and wind speed at the Ameriflux PRR site

The average temperature profile (in difference from the temperature at 16 m) is shown in Fig. 6a. The impact of wind speed on the surface layer temperature profile is clear. For $U_a < 2$ m s$^{-1}$, the temperature decreases rapidly all the way down to the surface and the Richardson number is overwhelmingly greater than 0.25 (Fig. 6b), which is the traditionally cited limit value



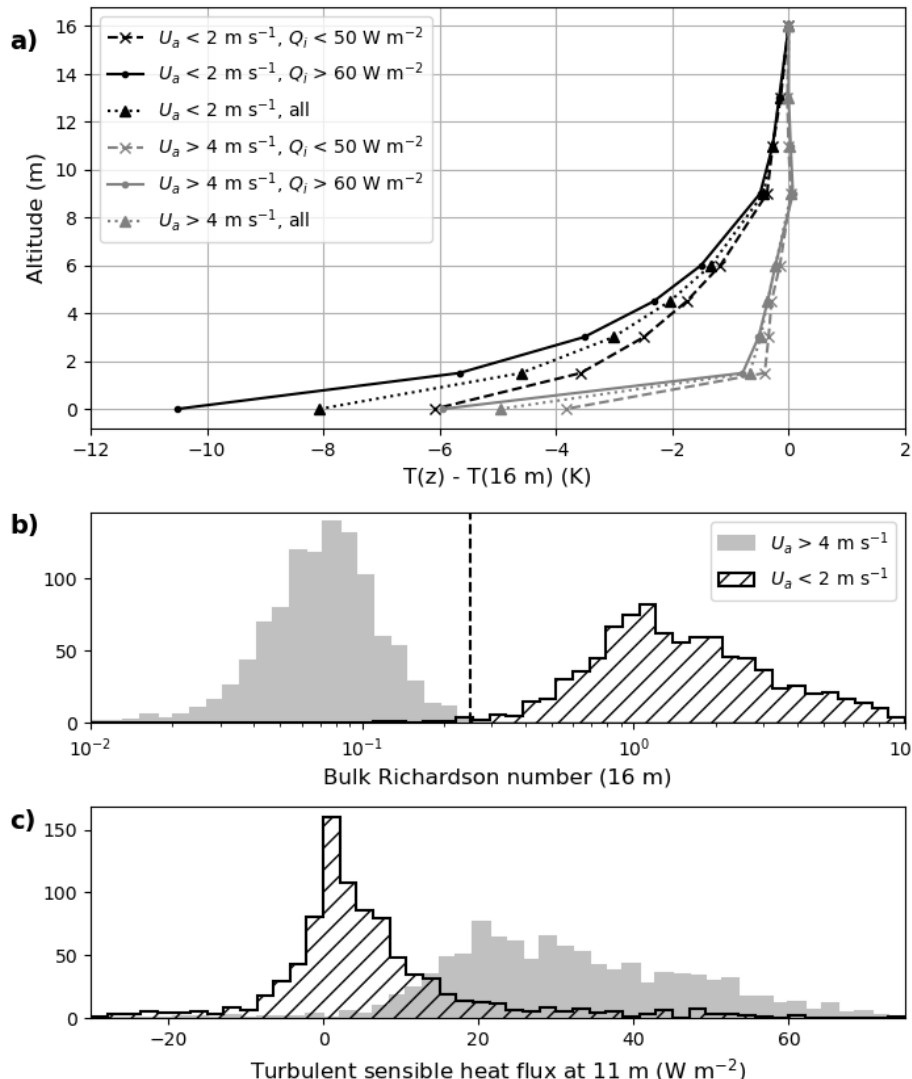

**Figure 6.** Panel a: average temperature difference from 16 m for wind speeds at 16 m smaller than 2 m s$^{-1}$ (black lines) and higher than 4 m s$^{-1}$ (grey lines). Continuous lines correspond to $Q_i > 60$ W m$^{-2}$ and dashed lines to $Q_i < 50$ W m$^{-2}$. Panel b: histogram of $R_b$ values calculated at 16 m, for wind speeds greater than 4 m s$^{-1}$ (filled grey) or smaller than 2 m s$^{-1}$ (hashed black). Panel c: histogram of turbulent sensible heat flux measured at 11 m (identical colours).

beyond which turbulence collapses. However, while the turbulent sensible heat flux has a low mean value of 4 W m$^{-2}$, its distribution remains quite spread out, with 5th and 95th percentiles of -12 and 30 W m$^{-2}$ respectively (Fig. 6c). This indicates that there is some remaining turbulence.

For $U_a > 4$ m s$^{-1}$, the temperature gradient is very weak (approximately 0.5 °C) down to 1.5 m, with a strong temperature





gradient remaining in the last meters. The top to bottom temperature difference is nevertheless smaller than for $U_a < 2$ m s$^{-1}$,
leading to $R_b$ values that are smaller than 0.25. Accordingly, the turbulent sensible heat flux is much larger than for the lower
wind speeds: its mean is 32 W m$^{-2}$, with 90% of values between 11 and 60 W m$^{-2}$. The fact that both $R_b$ and the turbulent
sensible heat flux have clearly distinct distributions for wind speeds greater than 4 and lower than 2 m s$^{-1}$ suggests that a
threshold wind speed for sustainable turbulence probably occurs in this range. It should further be noted that while only the
bulk Richardson number at 16 m is calculated here, the distributions are similar at other altitudes higher than 6 m. The impact
of the radiative input ($Q_i$) is also clear in Fig. 6a. The average profiles corresponding to values of $Q_i > 60$ W m$^{-2}$ exhibit a
larger temperature gradient than those corresponding to values lower than 50 W m$^{-2}$, especially at low wind speeds. This is
coherent with Sect. 2: greater radiative cooling leads to a larger SBI.

The relationship between the average air to surface temperature difference and wind speed is shown in Fig. 7c. $\Delta T_{as} = T_a - T_s$ decreases with $U_a$, reaching a minimum for $U_a > 5$ m s$^{-1}$, and there is a clear distinction between the averages
corresponding to $Q_i$ lower than 50 and greater than 60 W m$^{-2}$ respectively. $\Delta T_{as}$ can further be broken down into $\Delta T_{cs} = T_{1.5m} - T_s$ (Fig. 7b) and $\Delta T_{ac} = T_a - T_{1.5m}$ (Fig. 7a). $\Delta T_{ac}$ exhibits a very clear 'S' shape, collapsing to less than 1 K at wind
speeds higher than 4 m s$^{-1}$. $\Delta T_{cs}$, on the other hand, is maximum around 3 m s$^{-1}$ for both ranges of $Q_i$. These behaviours are
reminiscent of the two-layer model (Sect. 2.1): the main difference here is that $\Delta T_{cs}$ remains positive instead of decreasing to
negative values at low wind speeds.

Examination of the temperature profiles and gradients in relation to wind speed at the PRR site therefore suggests that a
2-layer model may be able to reproduce the temperature gradients, with the temperature at 1.5 m being a proxy for the canopy
temperature. The observations are compared in more detail to the models in Sect. 4.2.

## 4.2 Evaluation of WRF SEB-SL model compared to the PRR site measurements

The output of the 1-layer models (oMYJ and mMYJ) are shown in blue in Fig. 7c. Both tend to similar values as the observations
for low wind speeds, although oMYJ does not reach a constant regime because $\zeta$ is limited to values of 1. Because this limit is
removed in mMYJ, it better reproduces two regimes separated by a transition; this transition is however more gradual because
the PRR stability function was used. Both models, however, predict too small values of $\Delta T_{as}$ at high wind speeds compared
to the observations.

The 2-layer models, on the other hand, both show a much more gradual decrease of $\Delta T_{as}$ with $U_a$. Indeed, the decrease is
so gradual in the output of oMP that it is not possible to discern two distinct regimes - even though the stability function used
is Businger-Dyer, which is very short-tailed (see Fig. 3a, black lines compared to the grey line which corresponds to the PRR
stability function). One reason for this is that many limits are placed to maintain turbulence: $u^*$ cannot become larger than 0.07
m s$^{-1}$, $\zeta$ must remain smaller than 1, and when the wind speed is used for calculating the turbulent diffusion coefficient it takes
a minimum value of 1 m s$^{-1}$ (this is only the case within the surface layer modules, so that WRF still outputs wind speeds
values less than 1 m s$^{-1}$). Although it is true that some turbulence is always maintained, as shown by the measurements at the
PRR site (Fig. 6), the result is that Noah-MP model outputs too low $\Delta T_{as}$ values at very low wind speeds. oMP also does not



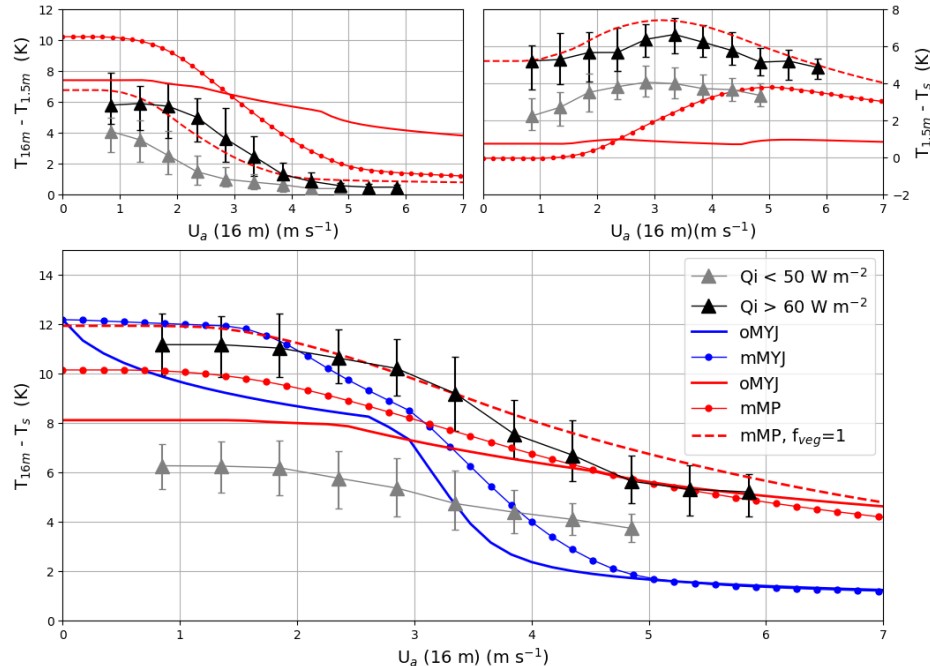

**Figure 7.** Panel a: average temperature difference between $z_a = 16$ m and 1.5 m ($\Delta T_{ac}$) as a function of wind speed at 16 m. Black and grey lines indicate averaged measurements for $Q_i > 60$ W m$^{-2}$ and $Q_i < 50$ W m$^{-2}$ respectively. Panel b: same, but for average temperature difference between 1.5 m and the surface ($\Delta T_{cs}$). Panel c: same, but for the temperature difference between 16 m and the surface ($\Delta T_{as}$). The blue continuous and dotted lines correspond to the output of the oMYJ and mMYJ models respectively, for input values $Q_i = 65$ W m$^{-2}$, $T_a = 263.15$ K, $T_g = 271.15$ K and $\Lambda_s = 1$ W m$^{-2}$ K$^{-1}$. The red continuous and dotted lines correspond to the output of the oMP and mMP models respectively, with the same input values of $Q_i$, $T_a$, $T_g$ and $\Lambda_s$. The red dashed line corresponds to the same simulation as the dotted red line, except that f$_{veg} = 1$.

reproduce the individual behaviour of $\Delta T_{cs}$ and $\Delta T_{ac}$: its calculated $\Delta T_{ac}$ does not have an 'S' shape as a function of wind speed and $\Delta T_{cs}$ exhibits no maximum.

The behaviour of mMP is more satisfactory. $\Delta T_{ac}$ shows a clear transition between a low-wind speed, high gradient state
and a high-wind speed state where the gradient is close to 0. $\Delta T_{cs}$ has a maximum between 3 and 5 m s$^{-1}$ (depending on the value of f$_{veg}$). $\Delta T_{as}$, finally, is close to the observed value both in the high and low wind speed limits. Two things must be noted here. First, that values of $\Delta T_{cs}$ remain positive at low wind speeds because, as noted in Sect. 3.2, a constant $K$ equal to $5 \times 10^{-4}$ m s$^{-1}$ has been added to $C_{D,a}U_a$. Similar to the limits imposed in oMP, this serves to maintain a certain level of turbulence and avoid the collapse of the turbulent sensible heat flux. Without this, $\Delta T_{cs}$ would decrease much more strongly
as described in Sect. 2.1. Adding a constant, as opposed to imposing a maximum value, is a more gradual method which does not distort the shape of the transition. The constant value is chosen to best represent the observations, and should be discussed in regards to other datasets.



Secondly, two versions of mMP are shown in Fig. 7. The first corresponds to $f_{veg} = 0.3$ and $\epsilon_v = 0.5$, which are the values which would be calculated by WRF from a LAI of 0.73 according to Eq. 10. The second corresponds to $f_{veg} = 1$ and $\epsilon_v = $

0.15. The results are substantially different, especially as concerns the canopy temperature (and therefore $\Delta T_{cs}$ and $\Delta T_{ac}$). Indeed, as outlined in Sect. 2.1, the canopy tends to become colder than the surface for higher values of $\epsilon_v$, and this is the case for the simulation with $f_{veg} = 0.3$. Furthermore, the transition wind speed (for $\Delta T_{ac}$) and wind speed at maximum $\Delta T_{ac}$ is shifted to lower values for the simulation with $f_{veg} = 1$. These two sets of values both correspond to $\epsilon_c = 0.15$, and therefore to the same radiative flux balance. However, the difference in outcome suggests that due to the turbulent fluxes, separating a bare

from a vegetated fraction is not equivalent to considering only one layer, but with lower emissivity. The simulation with $f_{veg} =$ 1 seems to perform better. One possible explanation is linked to the size of the eddies transporting heat. If they are of similar size to the typical distance between the trees, the turbulent transport of heat would not necessarily behave differently over a "bare" fraction than over a "vegetated" fraction. Instead, all turbulent transport would occur in an averaged manner. Indeed, the turbulent characteristics calculated in Sect. 3.1 are likely representative of this average, depending on the instrument footprint.

At the PRR site, average tree distance is approximately 2.2 m assuming that the trees are homogeneously distributed (which seems reasonable from site photos, Fig. 4). It is, however, not clear whether this is a robust feature of 2-layer model. Indeed, oMP does not appear to perform better when $f_{veg}$ is set to 1 and $\epsilon_c$ to 0.15: its calculated values of $\Delta T_{as}$ decrease more rapidly with wind speed, and therefore remain smaller than the measured temperature difference over the whole wind speed range (not shown). In the following, the values of $f_{veg} = 0.3$ and $\epsilon_v = 0.5$ are used for both 2-layer models.

The models are then run over all the PRR measurement points. Compared to the above analysis, this makes it possible to evaluate their behaviour for a wide variety of input values. Overall, all models capture some of the variability in $\Delta T_{as}$, probably due to the influence of the downwards radiative fluxes. It is clear however that the 1-layer models always underestimate $\Delta T_{as}$ when the measured $\Delta T_{as}$ is lowest, which corresponds to conditions of high wind speeds (Fig. 8). oMYJ also underestimates when the measured $\Delta T_{as}$ is very high, but this effect has been corrected in mMYJ by allowing a stronger decrease in turbulence.

The root mean square error (RMSE) of mMYJ is therefore approximately 2.8 K as opposed to 3.4 K for the original MYJ scheme.

The 2-layer models both perform better than the 1-layer models, supporting the idea that they are more adapted for use in a forest environment. The original Noah-MP model cannot reproduce strong values of $\Delta T_{as}$ because of excessive forced turbulence; mMP fares better in that regard. Its RMSE is slightly better (2.2 instead of 2.3 K). Note that running mMP with

$f_{veg} = 1$ and $\epsilon_v = 0.15$ leads to an RMSE of 2.1 K.

## 5  Conclusions and perspectives

A simple 2-layer analytical model of the stable surface layer was developed and contrasted with the existing 1-layer models of van de Wiel et al. (2017). The 2-layer model predicted a more gradual dependency of $\Delta T_{as}$ on the wind speed than the 1-layer models with equivalent roughness lengths and stability function. The top layer exhibited the 'S' shape dependence of

the temperature gradient on the wind speed which is typical of 1-layer models. The bottom layer, on the other hand, had a



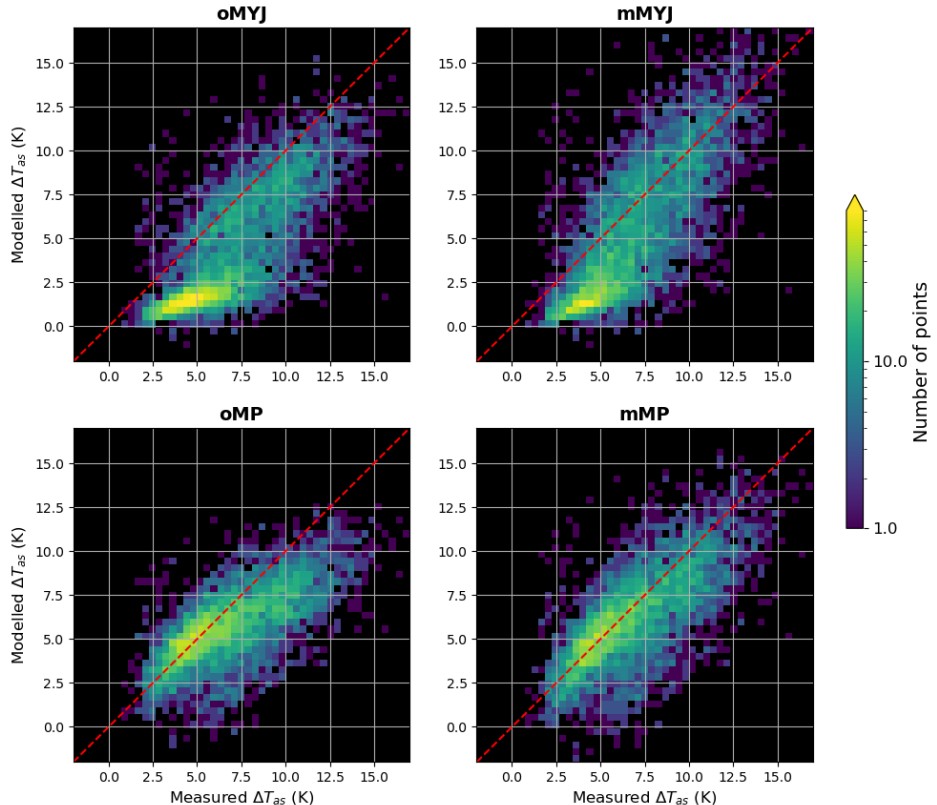

**Figure 8.** 2D histograms of the modelled vs measured $\Delta T_{as} = T_{16m} - T_s$ for the four models. Top row: 1-layer models (left: oMYJ and right: mMYJ). Bottom row: 2-layer models (left: oMP and right: mMP). The colour represents the number of points, in a log-normal scale and the red dashed line corresponds to the 1:1 line.

maximum temperature gradient at the transition wind speed. However, results depended strongly on the value of the first layer emissivity. Insights gained from the theoretical models were applied to study two surface layer/land surface model modules in WRF: Noah-MP and the Noah-MYJ combination. It was found that these models tend to set very restrictive boundaries on the turbulent diffusion coefficients and stability parameters, so that strong temperature gradients cannot be reached.

A combined approach was then used to study the performance of different surface layer models in more detail. First, an extensive set of measurements from the Poker Flats Research Range was analysed. It was found that under clear-sky, snow-covered, night-time conditions, the temperature gradient depended strongly on both the downwards longwave flux and the wind speed. When the wind speed at 16 m was smaller than 2 m s$^{-1}$, the temperature profile showed a very strong inversion down to the surface and the Richardson number was larger than 0.25, the traditional "cutoff" value for turbulence. Nevertheless,

some turbulent sensible heat flux remained. On the other hand, when the wind speed was larger than 4 m s$^{-1}$, the temperature profile was roughly constant down to 1.5 m, below which a strong temperature gradient remained. The Richardson number was then below 0.25, corresponding to the traditional "weakly stable" regime. Furthermore, the dependence of the individual layer



temperature inversion on wind speed were qualitatively similar to the theoretical 2-layer model.

Four different SEB-SL models were then coded into Python: the Noah-MYJ combination, Noah-MP, and modified versions of

the two. These were compared to the observations first qualitatively, and then by inputting measured values of temperature and wind speed at different altitudes in the surface layer and comparing the outputted value of $\Delta T_{as}$ to the measurements. It was found that the 2-layer models both gave better results than the 1-layer models, which tended to predict too low temperature gradients at high wind speeds. On the other hand, the original Noah-MP predicted too low temperature gradients at low wind speeds. All in all, the modified Noah-MP gave the best results, especially for the individual layer temperature gradients.

Open questions remain concerning the impact of local parameters on the simulations. Although the PRR site is classified as "Evergreen Needleleaf Forest", it trees are very short and spaced out, and its emissivity and roughness length low for a forest site. These parameters were shown to impact the behaviour of the lowest layer temperature gradient. Indeed, at high emissivities, the canopy layer is theoretically predicted to become colder than the surface. Furthermore, the value of the turbulent diffusion parameter for the surface to canopy air heat exchanges is taken from Monin–Obukhov similarity theory which

assumes a log wind profile. Other parametrisations, such as the log-exp profile of Mahat et al. (2013) which is implemented in Noah-MP, could conceivably yield better results in a denser forest. It would therefore be necessary to test the behaviour of the model compared to a denser forest site with higher trees.

The major perspective arising from the present paper is the implementation of the modified SEB-SL models into the main WRF framework. Once this is done, the impact of the modifications on model output can be tested over real cases. Because wind

speed can vary locally due to topography in the continental Arctic. The modifications suggested here are therefore expected to impact the spatial distribution of near-surface SBIs during winter anticyclonic episodes, with resulting consequences for the modelling of pollutant dispersion and pollution episodes. Another major question concerns the impact of clouds on the surface-layer stability and SBI, and how transitions between cloudy and clear surface layers are represented by WRF.

*Code and data availability.* The modified simplified versions, recoded in Python, of the Noah-MYJ and Noah-MP schemes from the WRF

4.5.1 meso-scale model and data from the Ameriflux Poker Flats Research Range (PRR) site used in this paper are permanently archived at https://doi.org/10.5281/zenodo.8347090.

# Appendix A: Conceptual 2-layer model development

The surface energy balance corresponding to the system in Fig. 1 is:

$$\mathrm{LW}_{d,bc} - \mathrm{LW}_{u,bc} + G + H_c = 0 \tag{A1}$$

where $\mathrm{LW}_{d,bc}$ and $\mathrm{LW}_{u,bc}$ are the downwards and upwards fluxes below the canopy level, G is the ground heat flux, and $H_c$ is the turbulent sensible heat exchange between the canopy and the surface. Each flux can then be parametrized.





$$\mathrm{LW}_{d,bc} = (1-\epsilon_c)\mathrm{LW}_d + \epsilon_c\sigma T_c^4$$

$$\mathrm{LW}_{u,bc} = \sigma T_s^4 \tag{A2}$$

$$\approx \sigma T_a^4 - 4\sigma T_a^3(T_a - T_s)$$

assuming $|T_a - T_s| \ll T_a$, with $\epsilon_c$ is the canopy emissivity, $T_c$ the canopy temperature, $T_a$ the air temperature above the canopy and $T_s$ the surface temperature. $\mathrm{LW}_d$ is the downwards longwave flux above the canopy level.

$$H_c = -\rho C_p C_{D,c} U_c(T_s - T_c) \tag{A3}$$

with $\rho$ the air density, $C_p$ the heat capacity of air, $U_c$ the canopy wind speed and $C_{D,c}$ the turbulent diffusion coefficient for the surface to canopy sensible heat exchange.

$$G = -\frac{\lambda_s}{d_s}(T_s - T_g) \tag{A4}$$

with $\lambda_s$ the snow conductivity, $d_s$ the snow depth and $T_g$ the ground temperature. The surface energy balance equation can
then be reorganised to Eq. 1:

$$-(1-\epsilon_c)Q_i - \Lambda_s(T_a - T_g) + \Delta T_{cs}\left[\rho C_p C_{D,c}U_c + 4\sigma T_a^3 + \Lambda_s\right]$$
$$+ \Delta T_{ac}\left[(1-\epsilon_c)4\sigma T_a^3 + \Lambda_s\right] \tag{A5}$$
$$= 0$$

Similarly, the energy balance applied to the canopy layer yields:

$$\mathrm{LW}_d - \mathrm{LW}_u - (\mathrm{LW}_{d,bc} - \mathrm{LW}_{u,bc}) + H_a - H_c = 0 \tag{A6}$$

where $\mathrm{LW}_u$ is the upwards longwave flux measured above the canopy level, $U_a$ is the air wind speed and $C_{D,a}$ the turbulent
diffusion coefficient for the canopy to air sensible heat exchange. Similarly, this transforms to:

$$-\epsilon_c Q_i - \Delta T_{cs}\left[\rho C_p C_{D,c}U_c + 4\epsilon_c\sigma T_a^3\right] + \Delta T_{ac}\left[\rho C_p C_{D,a}U_a + 4\epsilon_c\sigma T_a^3\right] = 0 \tag{A7}$$

Summing Eqs. 1 and A7 then yields Eq. 2:

$$-Q_i - \Lambda_s(T_a - T_g) + \Delta T_{cs}\left[(1-\epsilon_c)4\sigma T_a^3 + \Lambda_s\right]$$
$$+ \Delta T_{ac}\left[4\sigma T_a^3 + \Lambda_s + \rho C_p C_{D,a}U_a\right] \tag{A8}$$
$$= 0$$



*Author contributions.* JM developed and coded the models, performed the data treatment and analysis, and prepared the manuscript. FR and
JCR provided supervision, guidance and editing.

*Competing interests.* The authors declare that they have no conflict of interest.

*Acknowledgements.* Funding for the AmeriFlux data portal was provided by the U.S. Department of Energy Office of Science. The US-Prr
(PokerFlats Research Range) site is supported by JAMSTEC and IARC/UAF collaboration study (JICS) and Arctic Challenge for Sustain-
ability Project (ArCS, Sept 2015 - Mar 2020) and ArCSII (Jul 2020-). Computer modelling benefited from access to IDRIS HPC resources
(GENCI allocations A011017141 and A013017141) and the IPSL mesoscale computing centre.



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
