# Peer review of "Evaluation and development of surface layer scheme representation of temperature inversions over boreal forests in Arctic wintertime conditions"

_EGUsphere, 2023_

## Referee Comment (RC1)

**Comments on "Evaluation of WRF 4.5.1 surface layer scheme representation of temperature inversions over boreal forests"**

Sunday 29$^{\text{th}}$ October, 2023

**General:**

The manuscript by Maillard et al. evaluates the performance of two simplified surface layer schemes (extracted from the WRF model and modified) in reproducing the surface-based inversion over forest areas in the Arctic winter. The authors designed a simple two-layer analytical model to capture the temperature gradients (e.g., air-canopy, canopy-surface, and air-surface temperature differences), and then compared these conceptual models with modified simplified WRF surface-layer schemes to investigate the relationship between temperature gradient and wind speed based on a long-term *in situ* measurements. The modified models correct the limits on turbulence collapse under strong stable conditions to some extent. The paper provides some insights into the limitations of the common surface-layer schemes in WRF model and accordingly proposes to improve their performance in representing surface temperature inversions over boreal forests. Despite the paper offering good research idea and valuable insights, there are still some presentations require significant refinement to match the publication standards. Given the concerns, I recommend a major revision.

**Major comments:**

1. Conceptual model (section 2.1.1, section 2.1.2 & Appendix A).

Please double check and re-derive the conceptual model (Eqs.1,2,4,5) carefully. I have derived three times and found that the Eq.4 is incorrect and lacks coefficient $\Lambda_s$ for both $\Delta T_{ac}$ and $\Delta T_{cs}$. The correct expression should be:

$\Delta T_{ac} = \frac{\Lambda_s(T_a-T_g)+Q_i[1+\Lambda_s/(4\sigma T_a^3)]}{4\sigma T_a^3+2\Lambda_s}$ and $\Delta T_{cs} = \frac{\Lambda_s(T_a-T_g)-\Lambda_s Q_i/(4\sigma T_a^3)}{4\sigma T_a^3+2\Lambda_s}$

Despite it does not affect solving the total temperature gradient $\Delta T_{as}$, it has a large impact on $\Delta T_{ac}$ and $\Delta T_{cs}$, thus affecting the correctness of the results, as shown in Fig.2 and the relevant values in the text (e.g., Lines 122-123: $\Delta T_{ac}$ will becomes positive unless the snow depth is less than 1.8 cm when the lost coefficient $\Lambda_s$ is added, instead of 8 cm).

Equation (1) is correct, I derived the same form. However, I have derived several times and have not gain the Eq.2 in the text. The expression I derived is:

$-Q_i - \Lambda_s(T_a - T_g) + \Delta T_{ac}[4\sigma T_a^3 + \Lambda_s + \rho C_p C_{D,a}U_a] + \Delta T_{cs}\Lambda_s = 0$

i.e., the term of $\Delta T_{cs}$ does not contain $(1-\epsilon_c)4\sigma T_a^3$. Perhaps I was missing in some where.

2. Using only two short paragraphs (lines 365-375) to analyze the final validation of the modified SL models raises doubts about the reliability of the model. Therefore, it is advisable to provide a more comprehensive analysis and discussion of the results obtained using all "full measurements" as inputs to

the model (as shown in Figure 8). I recommend including a figure (similar to Fig.7) to present the final simulations based on the data used in Fig.8. This will strengthen the overall analysis and the reliability of your model.

3. The paper mentions the "S" shaped relationship between $\Delta T$ and $U$ many times but does not provide a visual representation of this relationship using raw observations. I'm curious what exactly does the raw observations look like? It would be beneficial to include this content in the text. Adding a scatter plot of $\Delta T$ versus $U$ using all the raw observations, with colors indicating the value of the stability parameter ($z/L$ or $Ri_b$), would help illustrate whether the stability regimes align with the wind speed divisions mentioned in the text. Additionally, it is important to explain why the criteria of $U_a < 2$ m/s and $U_a > 4$ m/s were used to separate stability regimes. Clarifying whether this choice is based on Figure 3c or other considerations would strengthen the methodology. Providing this context will enhance the understanding of how the stability regimes were defined and improve the robustness of the methods employed.

4. The paper does not provide a detailed discussion of the limitations of the conceptual model, which is based on a number of assumptions (e.g., ignoring shortwave radiation and latent heat flux, ...) and has its validation based on curated observations (filtering data with some criteria, e.g., snow depth $> 10$ cm, clear-sky, downward shortwave $< 30$ W/m$^{-2}$, etc.). It is important to consider whether the model would still perform effectively in all real situations if all raw observations were used as model inputs instead of filtered data? This issue should be considered, as the last paragraph mentions the modified models will be applied to the WRF framework (the real scenarios beyond the filtered observations).

**Minor comments:**

* The title of the paper could lead the readers into thinking that it is an evaluation of the performance of the SLM in the WRF model. You actually did not run WRF, so it is recommended to modify the title for more appropriate.

* Line 56, It's better to give the full name of the abbreviation LMDZ.

* Lines 81-83, Could you please explain why the derivation of Eq.1 ignores shortwave radiation and latent heat flux? Is it because this study focuses only on conditions during Arctic winter?

* Figure 1, It would be better to mark the energy balance equation like Eq.A1 at the interface between each two layers in Fig.1, which well help the reader understand the derivation of the formula in the texts. This is a recommendation only and is not mandatory.

* Line 109: "...the weakly and strongly stable limits.". How to define these two regimes?

* Caption of Figure 3: better "(a)" than "Panel a:"; "(b, c) Same as (a), but for $\Delta T_{cs}$ and $\Delta T_{as}$, respectively." than "Panel b: same, for $\Delta T_{cs}$. Panel c: same, for $\Delta T_{as}$". In addition, please keep figure labels consistent – it's better not to use "panel" for some figures and "a, b, c" for others.

* Figure 5c: Colorbar is missing. And, why did you choose to use a histogram instead of a scatter plot with raw data?

* Lines 245-250: The shape of $\psi$ of WRF looks similar to the measurements when $z/L < 20$, with the exception of the range $3 < z/L < 10$. So, the statement of "...more gradual than WRF function" appears to be inaccurate.

* Line 257: Fig.5a shows the fitting curve deviates significantly from the observations when $z/L > 3$. Questions that could be addressed include: What's the number of valid data points used for the fitting shown in the graph? Is the fitting function statistically significantly? What is the standard deviation of the fitting coefficients?

* Line 284: What am I missing? I did not see a comparison of the calculated turbulent diffusion coefficients with outputs from actual WRF runs. It is recommended that such a comparison be included, perhaps as supplementary material, to validate the modified model's performance is consistency with WRF runs.

* Line 310: How are the specific threshold values of 50 and 60 W/m$^2$ for $Q_i$ determined in results analysis (e.g., Fig.6 & Fig.7)? What is the rationale for using these values (50 and 60) for data grouping?

* Line 331: "see Fig. 5a" than "see Fig. 3a".

* Caption of Fig.7: Where the specific values for $T_a$, $T_g$ and $\Lambda_s$ are obtained from?

* Fig.7: the red "oMYJ" in the legend should be corrected to "oMP". Why does the "mMP" overestimate $\Delta T_{ac}$ under weak wind speed in Fig.7a?

* Line 433: What's expression for $H_a$? $-\rho C_p C_{Da} U_a (T_c - T_a)$?

* To enhance the clarity of the derivation process in the Appendix, it is recommended to include the missing details regarding the approximation $|T_a - T_s| << T_a$ and the use of the equation $T_s - T_g = \Delta T_{ag} - \Delta T_{ac} - \Delta T_{cs}$, and so on.

---

## Author Comment (AC1)

**RC :**

   The manuscript by Maillard et al. evaluates the performance of two simplified surface layer schemes (extracted from the WRF model and modified) in reproducing the surface-based inversion over forest areas in the Arctic winter. The authors designed a simple two-layer analytical model to capture the temperature gradients (e.g., air-canopy, canopy-surface, and air-surface temperature differences), and then compared these conceptual models with modified simplified WRF surface-layer schemes to investigate the relationship between temperature gradient and wind speed based on a long-term *in situ* measurements. The modified models correct the limits on turbulence collapse under strong stable conditions to some extent. The paper provides some insights into the limitations of the common surface-layer schemes in WRF model and accordingly proposes to improve their performance in representing surface temperature inversions over boreal forests. Despite the paper offering good research idea and valuable insights, there are still some presentations require significant refinement to match the publication standards. Given the concerns, I recommend a major revision.

**AC :** The authors would like to thank the Reviewer#1 for his/her careful review of our manuscript. We addressed each comment individually and have revised the manuscript accordingly.

**1) RC :**

   1. Conceptual model (section 2.1.1, section 2.1.2 & Appendix A).
   Please double check and re-derive the conceptual model (Eqs.1,2,4,5) carefully. I have derived three times and found that the Eq.4 is incorrect and lacks coefficient $\Lambda_s$ for both $\Delta T_{ac}$ and $\Delta T_{cs}$. The correct expression should be:
   $\Delta T_{ac} = \frac{\Lambda_s(T_a - T_g) + Q_i[1 + \Lambda_s/(4\sigma T_a^3)]}{4\sigma T_a^3 + 2\Lambda_s}$ and $\Delta T_{cs} = \frac{\Lambda_s(T_a - T_g) - \Lambda_s Q_i/(4\sigma T_a^3)}{4\sigma T_a^3 + 2\Lambda_s}$
   Despite it does not affect solving the total temperature gradient $\Delta T_{as}$, it has a large impact on $\Delta T_{ac}$ and $\Delta T_{cs}$, thus affecting the correctness of the results, as shown in Fig.2 and the relevant values in the text (e.g., Lines 122-123: $\Delta T_{ac}$ will becomes positive unless the snow depth is less than 1.8 cm when the lost coefficient $\Lambda_s$ is added, instead of 8 cm).
   Equation (1) is correct, I derived the same form. However, I have derived several times and have not gain the Eq.2 in the text. The expression I derived is:
   $-Q_i - \Lambda_s(T_a - T_g) + \Delta T_{ac}[4\sigma T_a^3 + \Lambda_s + \rho C_p C_{D,a} U_a] + \Delta T_{cs}\Lambda_s = 0$
   i.e., the term of $\Delta T_{cs}$ does not contain $(1 - \epsilon_c)4\sigma T_a^3$. Perhaps I was missing in some where.

**AC :** We disagree with Reviewer#1 and confirm that our Equation 2 was correct. We give here more details to derive Equation 2 from Equation (A6) :
Equation (A6) gives the energy balance applied to the canopy layer.
Here : $LW_d - LW_{d,bc} = \epsilon_c LW_d - \epsilon_c \sigma T_c^4 \approx \epsilon_c(LW_d + 4\sigma T_a^3 \Delta T_{ac} - \sigma T_a^4) \approx -\epsilon_c Qi + 4\epsilon_c \sigma T_a^3 \Delta T_{ac}$,
and : $LW_{u,bc} - LW_u = \epsilon_c \sigma(T_s^4 - T_c^4) \approx -4\epsilon_c \sigma T_a^3 \Delta T_{cs}$.
As a consequence, Equation (A6) yields :
$-\epsilon_c Q_i + 4\epsilon_c \sigma T_a^3 \Delta T_{ac} - 4\epsilon_c \sigma T_a^3 \Delta T_{cs} + \rho C_p C_{D,a} U_a \Delta T_{ac} - \rho C_p C_{D,c} U_c \Delta T_{cs} = 0$
which is equivalent to Equation (A7). Those equations have been added in the Appendix of the revised manuscript for the sake of clarity.
Equations 4 and 5 given by Reviewer#1 are correct. In the original manuscript, the value of $\Lambda_s$ had been set to 1 W m$^{-2}$ K$^{-1}$ when studying the asymptotic cases of Sect. 2.1.2. It was clarified in the caption of Figure 2. We have therefore explicitly indicated $\Lambda_s$ in Equations 4 and 5 and recalled that we used the value of $\Lambda_s = 1$ W m$^{-2}$ K$^{-1}$ for the asymptotic case. But Figure 2 and the associated discussion remain unchanged.

**2) RC :**

2. Using only two short paragraphs (lines 365-375) to analyze the final validation of the modified SL models raises doubts about the reliability of the model. Therefore, it is advisable to provide a more comprehensive analysis and discussion of the results obtained using all "full measurements" as inputs to the model (as shown in Figure 8). I recommend including a figure (similar to Fig.7) to present the final simulations based on the data used in Fig.8. This will strengthen the overall analysis and the reliability of your model.

**AC :** We thank Reviewer#1 for this question, which helps to improve the main conclusions of the paper. To gain insight about the reliability of the model, we have selected all available observations and binned them according to their wind speed $U_a$ values in intervals of width $0.5 \text{ m s}^{-1}$. This eliminates assumptions regarding input parameters such as the net radiation at the surface. Results are shown in Fig. 9.

It clearly shows that MYJ, whether in its original (oMYJ) or its modified (mMYJ) versions, repro-

[Figure]

**FIGURE 1** (a) Median temperature difference between $z_a = 16$ m and 1.5 m ($\Delta T_{ac}$) as a function of wind speed at 16 m. Black line indicates measurements binned them according to their wind speed $U_a$ values in intervals of width $0.5 \text{ m s}^{-1}$. (b,c) Same as (a), but for $\Delta T_{cs}$ and $\Delta T_{as}$, respectively. The blue continuous and dotted lines correspond to the output of the oMYJ and mMYJ models respectively. The red continuous and dotted lines correspond to the output of the oMP and mMP models respectively. The red dashed line corresponds to the same simulation as the dotted red line, except that $f_{veg} = 1$. The error bars on the measured or modelled values represent the interquartile range ($25^{\text{th}}$ and $75^{\text{th}}$ percentiles).

duces a too sharp transition due to the fact that it only considers a single layer, and strongly differs from the observations when $U_a$ values become larger than $2.5 \text{ m s}^{-1}$. mMYJ is in better agreement with the observations when the wind speed is weaker because the modelled $\Delta T_{as}$ values are obtained in a constant regime and enhanced by 2 K. This is the consequence of removing the limitation of $\zeta$ values to 1.

Regarding the 2-layer models, oMP slightly underestimates the strength of the inversion for small values of the wind speed $U_a$, even though the results are not too far from the error bars : the interquartile intervals barely overlap with those of the observed values. On the other hand, it appears obvious that it is actually due to compensation errors on the two layers taken individually : $\Delta T_{ac}$ is overestimated

while $\Delta T_{cs}$ is underpredicted.

The two versions of mMP provide by far the best results compared to the observations, especially when $f_{veg} = 1$. It captures the dependency of the two individual layers (atmosphere-canopy and canopy-surface) on the wind speed well.

This discussion has been added into the revised manuscript.

**3) RC :**

3. The paper mentions the "S" shaped relationship between $\Delta T$ and $U$ many times but does not provide a visual representation of this relationship using raw observations. I'm curious what exactly does the raw observations look like? It would be beneficial to include this content in the text. Adding a scatter plot of $\Delta T$ versus $U$ using all the raw observations, with colors indicating the value of the stability parameter ($z/L$ or $Ri_b$), would help illustrate whether the stability regimes align with the wind speed divisions mentioned in the text. Additionally, it is important to explain why the criteria of $U_a < 2$ m/s and $U_a > 4$ m/s were used to separate stability regimes. Clarifying whether this choice is based on Figure 3c or other considerations would strengthen the methodology. Providing this context will enhance the understanding of how the stability regimes were defined and improve the robustness of the methods employed.

**AC :** A real "inverted S" shaped relationship between $\Delta T$ and $U$ has been reported by VAN DE WIEL et al. (2017) based on measurements in Antarctica. The authors clearly noted a range of wind speeds where the temperature inversion was either strong or weak, leading to hysteresis phenomena. VAN DE WIEL et al. (2017) also observed this $S$ in their 1-layer conceptual models.

The observation of such an "inverted S" is more tricky in our study, because a key result in this paper is that the transition between the two modes is much more gradual in 2-layer models. This phenomenon corresponds to observations in a context where the surface is covered with trees. Over a forest surface or in multi-layer models, the "S" becomes blurred : the transition between the two modes is much more gradual than a true "S", as highlighted in Fig. 3c.

To clarify this, we have removed the references to the "S" in the revised manuscript, replacing by "transition" in $\Delta T_{as}$, and clarified the fact that the presence of trees attenuates the "S" shape. This is the reason why 2-layer models perform better.

The distinction between $U_a < 2$ m/s and $U_a > 4$ m/s is only used in Fig. 6b. It only serves to illustrate the importance of the wind speed for the wind regime stability. This separation is based on the distribution of the bulk Richardson number. Indeed, 65% of the data with $U_a > 4$ m/s have a $R_i$ lower than $R_{ic} = 0.25$. And 99.2% of the data with $U_a < 2$ m/s have a $R_i$ larger than $R_{ic}$. The two modes are therefore clearly separated with a negligible overlap. The purpose of this distinction in two modes is not to explain theoretically at which value of the wind speed there would be a transition, but simply that such a wind speed exists. At this specific site, the value of $U_a = 3 \, \mathrm{m\,s^{-1}}$ is quite plausible (see Fig. 7a). It is therefore important to note that those specific values $U_a < 2$ m/s and $U_a > 4$ m/s are not part of our methodology.

**4) RC :**

4. The paper does not provide a detailed discussion of the limitations of the conceptual model, which is based on a number of assumptions (e.g., ignoring shortwave radiation and latent heat flux, ...) and has its validation based on curated observations (filtering data with some criteria, e.g., snow depth $> 10$ cm, clear-sky, downward shortwave $< 30$ W/m$^{-2}$, etc.). It is important to consider whether the model would still perform effectively in all real situations if all raw observations were used as model inputs instead of filtered data? This issue should be considered, as the last paragraph mentions the modified models will be applied to the WRF framework (the real scenarios beyond the filtered observations).

**AC :** We agree with Reviewer#1 and have clarified the aim the paper. The goal is to model inversions in an Arctic winter context, hence the conditions on the absence of shortwave radiation, latent heat flux and the significant presence of snow on the ground. The assumptions used to filter the data help to select only situations in the Arctic winter. Apart from these conditions, the original models are not modified. We have modified the title of the article to include "in the Arctic winter" for the sake of clarity. The implementation of our modifications in WRF should be followed by a testing phase to find out how the model performs outside these restrictive conditions. This is however beyond the scope of this paper.

« This study focuses on the clear-sky surface layer in an Arctic winter context. Clear-sky periods have been identified as those when the net longwave radiation was less than $-30 \, \mathrm{W\,m^{-2}}$ (GRAHAM et al., 2017; MAILLARD et al., 2021). Wintertime conditions have been selected on periods between November and March when the downwelling shortwave radiation was lower the $30 \, \mathrm{W\,m^{-2}}$, the latent heat flux less than $5 \, \mathrm{W\,m^{-2}}$ and the snow depth greater than 10 cm. Outside these conditions, the original conceptual models have not been modified. The implementation of conceptual model improvements in WRF should be followed by a testing phase to find out how the mesoscale model performs outside these restrictive conditions. »

5) Minor comments :

— **RC :** The title of the paper could lead the readers into thinking that it is an evaluation of the performance of the SLM in the WRF model. You actually did not run WRF, so it is recommended to modify the title for more appropriate.

   **AC :** This is right. We have changed the title : "Evaluation and development of surface layer scheme representation of temperature inversions over boreal forests in Arctic wintertime conditions".

— **RC :** Line 56, It's better to give the full name of the abbreviation LMDZ.

   **AC :** The acronym of the LMDZ model stands for "Laboratoire de Météorologie Dynamique - Zoom" model. It has been detailed in the text.

— **RC :** Lines 81-83, Could you please explain why the derivation of Eq.1 ignores shortwave radiation and latent heat flux ? Is it because this study focuses only on conditions during Arctic winter ?

   **AC :** Yes, the study focuses on the clear-sky surface layer in an Arctic winter context. It has been mentioned before the derivation of Eq.1 and also clarified in the title.

— **RC :** Figure 1, It would be better to mark the energy balance equation like Eq.A1 at the interface between each two layers in Fig.1, which well help the reader understand the derivation of the formula in the texts. This is a recommendation only and is not mandatory.

   **AC :** Only two equations (A1 and A6) could fit in this figure because they correspond to the two energy balance equations, at the surface and in the canopy layer. We thank you for your advice but, as we do not want to overload the figure, we have decided not to take these suggestions into account.

— **RC :** Line 109 : "...the weakly and strongly stable limits.". How to define these two regimes ?

   **AC :** In this subsection, we do not define the two regimes (it is rather detailed in Fig. 6). Here we only consider the two asymptotic cases of the weakly and strongly regimes. The associated limits are defined by $U_a \to 0$ and $U_a \to \infty$ for the strongly and weakly regimes, respectively, while keeping $\Delta T_{as} > 0$. It has been added in Sect. 2.1.2.

— **RC :** Caption of Figure 3 : better "(a)" than "Panel a :"; "(b, c) Same as (a), but for $\Delta T_{cs}$ and $\Delta T_{as}$, respectively." than "Panel b : same, for $\Delta T_{cs}$. Panel c : same, for $\Delta T_{as}$". In addition, please keep figure labels consistent – it's better not to use "panel" for some figures and "a, b, c" for others.

   **AC :** Thank you for this comment. Labels have been made consistent in all figures.

— **RC :** Figure 5c : Colorbar is missing. And, why did you choose to use a histogram instead of a scatter plot with raw data ?

**AC :** We have added a colorbar on this figure. A bidimensional histogram is more appropriate here as we are interested in the density of points following the slope. It is clear that most points are well aligned, enabling the retrieval of average emissivity of the canopy layer.

— **RC :** Lines 245-250 : The shape of $\psi$ of WRF looks similar to the measurements when $z/L < 20$, with the exception of the range $3 < z/L < 10$. So, the statement of "...more gradual than WRF function" appears to be inaccurate.

**AC :** The expression "more gradual" was referring to the slope of the $\Psi$ function. We modified the sentence by the expression "more long-tailed".

— **RC :** Line 257 : Fig.5a shows the fitting curve deviates significantly from the observations when $z/L > 3$. Questions that could be addressed include : What's the number of valid data points used for the fitting shown in the graph ? Is the fitting function statistically significantly ? What is the standard deviation of the fitting coefficients ?

**AC :** The aim of the function was to reproduce the measurements over the zone of transition, i.e. approximately between $\zeta = 0.1$ and $\zeta = 1$. Indeed, the specific values of $\Psi$ are less important, in terms of modelling, for high values of $z/L > 3$, because at this point the turbulent heat flux will tend to collapse anyway : this is why the deviation at $z/L > 3$ was considered less problematic. Furthermore, the intermediate zone also exhibited a marked difference between the Businger-Dyer/WRF functions and the measurements.
The function was therefore fitted on this intermediate zone for the different heights ($z = 7.5$, 9, 11, 13 and 16 m), yielding coefficients which varied between in the ranges $[-5.5, -4.5]$, $[-1, 2]$ and $[10, 40]$ respectively (depending on the specific zone considered and the height). Plotting the function with these different parameters revealed little difference in behaviour over this range (see figure below for example). The values of $-5$, 0.1 and 20 were then rather arbitrarily chosen.

[Figure]

— **RC :** Line 284 : What am I missing ? I did not see a comparison of the calculated turbulent diffusion coefficients with outputs from actual WRF runs. It is recommended that such a comparison be included, perhaps as supplementary material, to validate the modified model's performance is consistency with WRF runs.

**AC :** We apologize, this sentence was confusing. It has been removed in the new version of the manuscript.

In addition, a evaluation of model's performance has been extended with the inclusion of Fig. 9 and the corresponding comments.

— **RC :** Line 310 : How are the specific threshold values of 50 and 60 W/m$^2$ for $Q_i$ determined in results analysis (e.g., Fig.6 & Fig.7) ? What is the rationale for using these values (50 and 60) for data grouping ?

**AC :** Equations (1) and (2) showed that the temperature differences $\Delta T_{ac}$ and $\Delta T_{cs}$ are functions of the isothermal net radiation $Q_i$. Plotting all temperature differences from 16 m (Fig. 6a and Fig. 7c) would be very confusing because of a large number of $Q_i$ values. We have rather chosen to gather all the observed values in two groups that do not overlap and are delimited by their values of $Q_i$. Threshold of 50 and 60 W m$^{-2}$ hence provide a clear view of the impact of $Q_i$ on the temperature profiles as a function of the wind speed. We have to note that those values are only used for illustrative purposes; the thresholds of 50 and 60 W m$^{-2}$ are not parameters of our methodology.

— **RC :** Line 331 : "see Fig. 5a" than "see Fig. 3a".

**AC :** Actually, it was Fig. 3a. It uses the Businger-Dyer stability function shown in Fig. 5a, but the description given here corresponds to Fig. 3a.

— **RC :** Caption of Fig.7 : Where the specific values for $T_a$, $T_g$ and $\Lambda_s$ are obtained from ?

**AC :** Specific values used in Fig. 7 are the same as those used for the theoretical model when studying the asymptotic cases in Sect. 2.1.2. It has been added in the new manuscript.

— **RC :** Fig.7 : the red "oMYJ" in the legend should be corrected to "oMP". Why does the "mMP" overestimate $\Delta T_{ac}$ under weak wind speed in Fig.7a ?

**AC :** Thank you for pointing this mistake. It has been corrected.
The overestimation of $\Delta T_{ac}$ under weak wind speed by the mMP model in Fig. 7 should not be interpreted as a model error and a discrepancy. In Fig. 7, models are run for specific input values of $Q_i$, $T_a$, $T_g$ and $\Lambda_s$, that do not necessarily correspond to the observed average values of the two groups represented by $Q_i < 50$ or $Q_i > 60$ W m$^{-2}$. A direct model to observations comparaison is now presented in the new Fig. 9. It highlights that the mMP model performs very well for all values of the wind speed and within the two layers.

— **RC :** Line 433 : What's expression for $H_a$ ? $-\rho C_p D_{D,a} U_a (T_c - T_a)$ ?

**AC :** Yes, $H_a = -\rho C_p D_{D,a} U_a (T_c - T_a)$. It has been added in the appendix.

— **RC :** To enhance the clarity of the derivation process in the Appendix, it is recommended to include the missing details regarding the approximation $|T_a - T_s| \ll T_a$ and the use of the equation $T_s - T_g = \Delta T_{ag} - \Delta T_{ac} - \Delta T_{cs}$, and so on.

**AC :** We have added some details in the Appendix :

$T_c$ can be written $T_c = T_a + \delta$ where $\delta = T_c - T_a$. Hence $T_c^4 = T_a^4 \left(1 + \dfrac{\delta}{T_a}\right)^4$ with $\dfrac{\delta}{T_a} \ll 1$. A first

order Taylor expansion leads to $T_c^4 \approx T_a^4 \left(1 + 4\dfrac{\delta}{T_a}\right) \approx T_a^4 + 4 T_a^3 (T_c - T_a)$.

Similarly $T_s^4 \approx T_a^4 + 4 T_a^3 (T_s - T_a)$.
We also added $G = \Lambda_s (T_g - T_a + \Delta T_{ac} + \Delta T_{cs})$.

---

## Author Comment (AC2)

**RC :** The authors took two surface layer schemes (Noah-MYJ and Noah-MP) out of WRF, and simplified them as stand-alone modules to evaluate their performance for temperature inversions over forests in the Arctic winter. Additionally, a conceptual model was also developed to investigate the impact of individual variables. To correct the limits of the WRF schemes on turbulent collapse, some modifications were inserted. The research provides some ideas on improving the surface layer models, especially under stable conditions. However, I found the structure of the manuscript is not well organized. The interpretation of the results is limited too. So, I would like to recommend a major revision before it can be published in this journal.

**AC :** The authors would like to thank the Reviewer#2 for his/her careful review of our manuscript. We addressed each comment individually and have revised the manuscript accordingly.

**1)    RC :** The title is misleading, since you didn't evaluate the surface layer schemes inside WRF. Especially, what's the difference of the surface layer models in the version 4.5.1 compared to the previous versions ?

**AC :** We agree that the title probably wasn't the most appropriate. We have modified it to "Evaluation and development of surface layer scheme representation of temperature inversions over boreal forests in Arctic wintertime conditions" to avoid confusion. Indeed, the goal of the paper wasn't to compare the surface layer models in this version of WRF compared to previous versions, but to compare their performance, specifically in wintertime Arctic conditions, and suggest improvements.

**2)    RC :** In section 2 the authors first introduce the conceptual model, then the two schemes from WRF, and then in section 3 the modified schemes are described after the measurements. I feel this organization is not straightforward and confusing. The connection between the conceptual model and WRF schemes is not clear.

**AC :** The aim of the conceptual model is to gain insight into the behaviour of a 2-layer surface layer model. By calculating the strongly and weakly stable limits, the differences with a 1-layer model are put forward and the comparison with the measurement data is clearer.

We agree with the reviewer that the current organization was confusing. We have re-organised the article in the following manner :

2. Conceptual model (previous part 2.1)

3. Measurements at the Ameriflux Poker Flats Research Range (previously part 3.1 et 4.1)

4. Description of and suggested modifications to the WRF surface layer models (previously parts 2.2 et 3.2)

5. Results (previously part 4.2)

**3)    RC :** The validation of the models (Figure 7) is based on a lot of input parameters regarded as constant values; however, these parameters should change with time. It's difficult to draw the conclusion that the modifications improve the model performance. This part should be expanded to gain more confidence.

**AC :** It is true that in Figure 7, most input parameters are set as constant values, for illustrative purposes. To gain insight about the reliability of the model, we have selected all available observations and binned them according to their wind speed $U_a$ values in intervals of width 0.5 m s$^{-1}$. This eliminates assumptions regarding input parameters such as the net radiation at the surface. Results are shown in Fig. 9.

 It clearly shows that MYJ, whether in its original (oMYJ) or its modified (mMYJ) versions, reproduces a too sharp transition due to the fact that it only considers a single layer, and strongly differs

[Figure]

**Figure 1** (a) Median temperature difference between $z_a = 16$ m and 1.5 m ($\Delta T_{ac}$) as a function of wind speed at 16 m. Black line indicates measurements binned them according to their wind speed $U_a$ values in intervals of width $0.5$ m s$^{-1}$. (b,c) Same as (a), but for $\Delta T_{cs}$ and $\Delta T_{as}$, respectively. The blue continuous and dotted lines correspond to the output of the oMYJ and mMYJ models respectively. The red continuous and dotted lines correspond to the output of the oMP and mMP models respectively. The red dashed line corresponds to the same simulation as the dotted red line, except that $f_{veg} = 1$. The error bars on the measured or modelled values represent the interquartile range ($25^{\text{th}}$ and $75^{\text{th}}$ percentiles).

from the observations when $U_a$ values become larger than 2.5 m s$^{-1}$. mMYJ is in better agreement with the observations when the wind speed is weaker because the modelled $\Delta T_{as}$ values are obtained in a constant regime and enhanced by 2 K. This is the consequence of removing the limitation of $\zeta$ values to 1.

Regarding the 2-layer models, oMP slightly underestimates the strength of the inversion for small values of the wind speed $U_a$, even though the results are not too far from the error bars : the interquartile intervals barely overlap with those of the observed values. On the other hand, it appears obvious that it is actually due to compensation errors on the two layers taken individually : $\Delta T_{ac}$ is overestimated while $\Delta T_{cs}$ is underpredicted.

The two versions of mMP provide by far the best results compared to the observations, especially when $f_{veg} = 1$. It captures the dependency of the two individual layers (atmosphere-canopy and canopy-surface) on the wind speed well.

Furthermore, Fig. 8 shows the performance of the models over all PRR site data (curated as explained in Sect.3), using the actual measurements as input parameters. The modified versions both perform better than the original in reproducing the temperature gradient, as evidenced by a more even distribution around the 1 :1 and the lower RMSE.

This discussion has been added into the revised manuscript.

**4) RC :** The investigations on the model results are limited, especially the last part with all measurements input. The discussion should be extended.

**AC :** We are of course open to suggestions on how to improve our discussion on the results, however it is unclear to us at the moment what else we could include to extend it. The model output was analysed and the RMSE calculated over several years of measurements (Sect. 6.2), showing that the modified versions better reproduce the temperature gradient as the original. Furthermore, the behaviour of

the temperature gradient in the individual layers was shown (Sect. 6.1) and the reasons for the diffe-rences in behaviour between the models are discussed.The behaviour of the modified models in other conditions (for example, other forest covers, or with shortwave radiation) is still an open question, as mentioned in the conclusion, however we consider it to be outside the scope of the present paper.

5) Minor comments :

— **RC :** The full name of WRF should be mentioned somewhere.

   **AC :** True. It has been added as the first occurrence of WRF.

— **RC :** Line 56 : LMDZ model should be explained.

   **AC :** The acronym of the LMDZ model stands for "Laboratoire de Météorologie Dynamique - Zoom" model. It has been detailed in the text.

— **RC :** The language needs to be improved. There are some spelling and grammar mistakes. For example, line 227 & 404 : a comma should be inserted before which. Line 401 is confusing.

   **AC :** The language has been checked carefully.
   L227 : "which" has been replaced by "that", which does not require a comma.
   L404 : A comma has been added.
   L401 has been rephrased : « Although the PRR site is classified as "Evergreen Needleleaf Forest" by the MODIS land-use categories, its characteristics are actually rather similar to a Wooded or Mixed Tundra : its trees are indeed very short and spaced out and its emissivity and roughness length are quite low for a forest site. »